# Antimicrobial and antibiofilm evaluation of thymol, sodium azide, and sodium lauryl sulfate against multidrug-resistant pathogens: An integrated experimental and computational study

Kamal A. Qureshi[ORCID][1]*, Adil Parvez[2]

**1** Department of Pharmaceutics, College of Pharmacy, Qassim University, Buraydah, Saudi Arabia,
**2** DSS Takara Bio India Pvt Ltd, New Delhi, India

* ka.qurishe@qu.edu.sa

## Abstract

Multidrug-resistant (MDR) pathogens represent a major global health challenge, underscoring the urgent need for new antimicrobial strategies that can effectively target both planktonic cells and biofilm-associated infections. This study integrated *in vitro* antimicrobial and antibiofilm assays with comprehensive *in silico* analyses to evaluate the repurposing potential of thymol (TM), sodium azide (SA), and sodium lauryl sulfate (SLS) against 17 bacterial and two fungal strains, including methicillin-resistant *Staphylococcus aureus* (MRSA) clinical isolates. TM showed the strongest overall antimicrobial activity, with low MIC/MBC values (0.10–0.20 mg/mL) and potent antibiofilm effects (MBIC: 0.20–0.39 mg/mL; MBEC: 0.39–0.78 mg/mL). In contrast, SA exhibited similar MICs (0.10–0.78 mg/mL) but required much higher concentrations for bactericidal and antibiofilm endpoints (MBC/MBIC/MBEC 6.25–100 mg/mL), whereas SLS displayed variable activity, with low MICs against most Gram-positive bacteria (0.10–0.20 mg/mL) but high MBC/MBIC/MBEC values (50–100 mg/mL), especially for Gram-negative biofilms. Molecular docking and 300 ns molecular dynamics (MD) simulations revealed that TM forms stable complexes with key microbial targets, most notably FtsZ ($\Delta G = -11.0$ kcal/mol; $K_d = 3.2 \times 10^{-10}$ M), supported by favorable MM/GBSA binding energies and restrained motions in principal component analysis/free-energy landscape (PCA/FEL) analyses. SA and SLS were primarily used as mechanistic comparators (respiratory inhibitors and membrane disruptors, respectively). In contrast, their non-ionic analogs, phenyl azide (PA) and lauryl sulfate (LS), were explored as potential scaffolds. LS showed a very high predicted affinity for UDP-3-O-acyl-*N*-acetylglucosamine deacetylase (LpxC) ($\Delta G = -19.9$ kcal/mol; $K_d = 1.7 \times 10^{-11}$ M), indicating promise for future optimization. *In silico* ADMET profiling identified TM as the most balanced candidate, combining broad-spectrum antibiofilm efficacy with a comparatively favorable predicted safety profile. Overall,

**Data availability statement:** All data supporting the findings of this study are publicly available. Processed molecular dynamics trajectories, analysis scripts, molecular docking results, MM/GBSA calculations, PCA and ADMET analysis files, and raw in vitro antimicrobial and antibiofilm assay data are available from Zenodo at DOI: https://doi.org/10.5281/zenodo.18427870.

**Funding:** Deanship of Graduate Studies and Scientific Research at Qassim University for financial support (QU-APC-2026).

**Competing interests:** The authors declare that they have no known competing financial interests or personal relationships that could have appeared to influence the work reported in this paper.

TM emerges as a viable repurposable antimicrobial agent, whereas LS-based derivatives represent computationally prioritized scaffolds that warrant further experimental validation.

---

## Introduction

### Global burden of antimicrobial resistance

Antimicrobial resistance (AMR) is one of the most urgent public health challenges of the 21st century, severely compromising the effectiveness of conventional antibiotics and complicating the management of infectious diseases [1,2]. In 2019, nearly 5 million deaths were associated with bacterial AMR, and projections suggest that this number may increase to 10 million annually by 2050 if current trends persist [1]. The rapid emergence and global dissemination of multidrug-resistant (MDR) pathogens, including methicillin-resistant *Staphylococcus aureus* (MRSA) and carbapenem-resistant *Enterobacteriaceae*, have rendered many frontline and last-resort antibiotics increasingly ineffective [2].

### Biofilm-associated infections and the need for new antimicrobials

Microbial biofilm formation exacerbates the AMR crisis. Biofilms are structured microbial communities embedded in a self-produced extracellular matrix that confers enhanced tolerance to antimicrobial agents and host immune responses [3,4]. Biofilm-associated infections are notoriously persistent and frequently lead to chronic infections, therapeutic failure, and relapse [5]. Although natural compounds have emerged as promising candidates for antibacterial and antibiofilm interventions, substantial gaps remain in understanding their molecular mechanisms, target specificity, and translational potential [6–9].

### Rationale for investigating Thymol (TM)

TM, a naturally occurring monoterpenoid phenol derived from *Thymus* species (thyme plants), has been reported to exhibit antibacterial, antifungal, and antibiofilm activities (Fig 1) [10,11]. Experimental studies have suggested that TM can disrupt microbial membranes, interfere with quorum-sensing pathways, reduce virulence factor expression, and inhibit biofilm formation [12,13]. Despite this growing body of evidence, several critical knowledge gaps persist. Specifically, TM has not been systematically evaluated across multiple essential microbial protein targets using long-timescale MD simulations; its activity has rarely been contextualized using mechanistic comparator compounds, and structurally related analogs with potentially improved drug-like properties remain underexplored. Addressing these gaps is essential for establishing a mechanistic and structural basis for TM's antimicrobial activity.

### Rationale for using mechanistic comparator compounds

To strengthen the mechanistic interpretation, sodium azide (SA) and sodium lauryl sulfate (SLS) were included in this study as mechanistic reference compounds rather

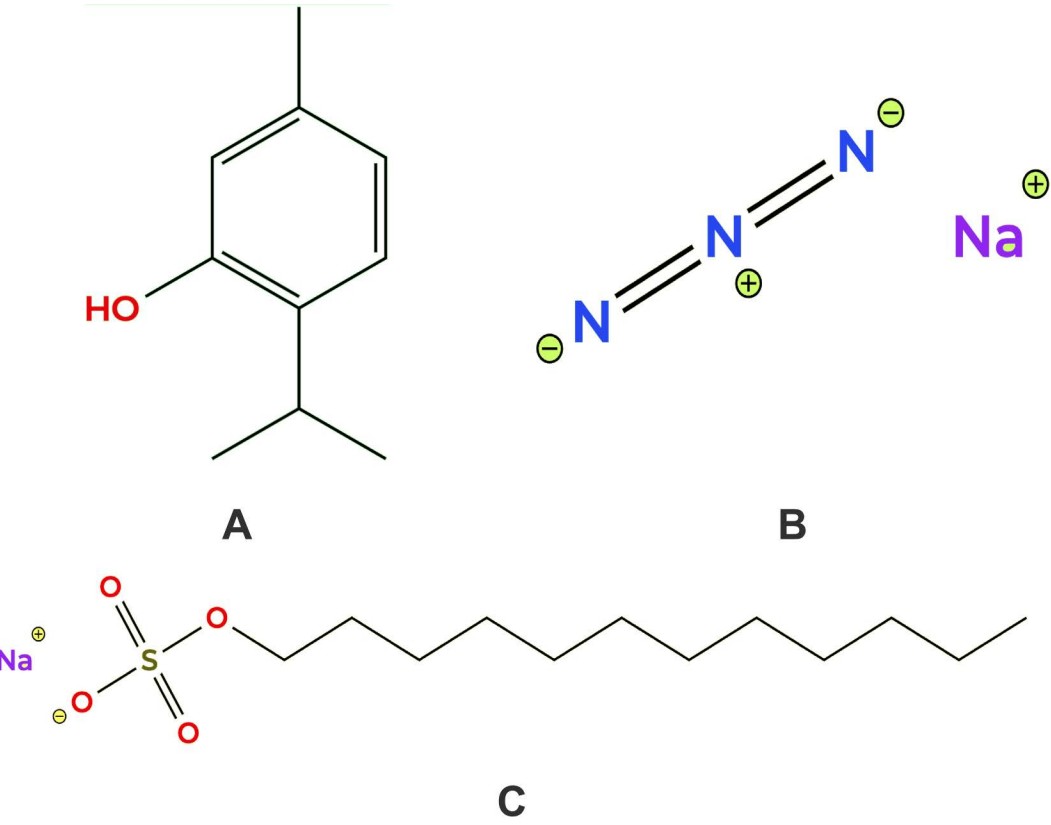

**Fig 1. Chemical structures of the tested compounds: (A) TM, (B) SA, and (C) SLS.**

than therapeutic candidates. SA is a well-characterized inhibitor of cytochrome oxidase that rapidly suppresses microbial respiration (Fig 1) [14,15], whereas SLS is an anionic surfactant that disrupts cellular membranes and induces lysis (Fig 1) [16,17]. These distinct, single-mechanism modes of action provide clear interpretive benchmarks against which TM's behavior can be compared. A conventional antibiotic was intentionally excluded, as the objective of this study was to distinguish mechanistic similarities and differences rather than to perform potency ranking against clinically optimized drugs.

## Structural analogs in computational exploration

Although SA and SLS are unsuitable for therapeutic use owing to toxicity and irritation concerns, their structural analogs may offer more drug-like scaffolds. Accordingly, phenyl azide (PA), an aromatic analog of SA, and lauryl sulfate (LS), a neutral analog of SLS, were incorporated into molecular docking and MD simulations. This approach enabled the exploration of whether simplified or modified structures exhibit enhanced binding stability, favorable physicochemical properties, or improved target interactions suitable for future optimization.

## Study hypotheses and objectives

Based on this rationale, we hypothesized that TM would demonstrate superior antimicrobial and antibiofilm activities relative to mechanistic comparators and form stable, biologically meaningful interactions with key microbial proteins. We further hypothesized that PA and LS may display improved interaction profiles, highlighting their potential as scaffolds for next-generation antimicrobial development. To test these hypotheses, this study employed an integrated workflow combining

antimicrobial susceptibility testing, biofilm inhibition assays, *in silico* ADMET profiling, molecular docking, 300 ns MD simulations, principal component analysis (PCA), free-energy landscape (FEL) mapping, and Molecular Mechanics Generalized Born Surface Area (MM/GBSA) calculations to achieve comprehensive phenotypic and mechanistic characterization.

## Materials and methods

### Test organisms

A total of 19 pathogenic microorganisms were used in this study, comprising 17 bacterial and two fungal strains. The bacterial panel included *Staphylococcus aureus* (*S. aureus*) ATCC 29213, *S. aureus* (clinical isolate; *S. aureus*-CI), methicillin-resistant *S. aureus* (MRSA-1 and MRSA-2), *Staphylococcus saprophyticus* (*S. saprophyticus*) ATCC 43867, *Staphylococcus epidermidis* (*S. epidermidis*) ATCC 12228, *Streptococcus pyogenes* (*S. pyogenes;* Group A) ATCC 19615, *Streptococcus pneumoniae* (*S. pneumoniae*) ATCC 49619, *Enterococcus faecalis* (*E. faecalis*) ATCC 29212, *Bacillus cereus* (*B. cereus*) ATCC 10876, *Escherichia coli* (*E. coli*) ATCC 25922, *Klebsiella pneumoniae (K. pneumoniae)* ATCC 27736, *Pseudomonas aeruginosa* (*P. aeruginosa*) ATCC 9027, *Salmonella typhimurium* (*S. typhimurium*) ATCC 13311, *Shigella flexneri* (*S. flexneri*) ATCC 12022, *Proteus vulgaris* (*P. vulgaris*) ATCC 6380, and *Proteus mirabilis* (*P. mirabilis*) ATCC 29906. The fungal strains used were *Candida albicans* (*C. albicans*) ATCC 10231 and *Aspergillus niger* (*A. niger*) ATCC 6275.

All ATCC reference strains were procured from Microbiologics® (St. Cloud, MN, USA). The non-ATCC microbial strains were obtained from the microbiology laboratory at a multi-speciality hospital, from Unaizah in Saudi Arabia. They were provided as archived, de-identified culture stocks that were previously collected for routine diagnostic purposes. No patient information or identifiers were shared, and no new samples were collected for this study. Therefore, no human subjects were involved, and ethical approval was not required under the institution's guidelines.

### Chemicals and reagents

The test compounds TM (CAS No. 89-83-8), SA (CAS No. 26628-22-8), and SLS (CAS No. 151-21-3) were procured from a certified supplier (Loba Chemie Pvt. Ltd., Mumbai, India). The specified purities of these compounds were as follows: SA, 99.5% w/w; TM, 99.0–101.0% w/w; and SLS, 91.0–92.0% w/w. This specification allows for slight analytical variances, particularly with TM, where the purity can exceed 100% owing to calibration deviations or methodological variations in testing. Unless otherwise specified, the remaining chemicals were procured from registered vendors (Sigma-Aldrich, USA; Oxoid Ltd., UK; Loba Chemie Pvt. Ltd., India).

### Antimicrobial profiles of TM, SA, and SLS

**Preliminary antimicrobial activity.** The preliminary antimicrobial activities of the test compounds (TM, SA, and SLS) were evaluated using the standard disc diffusion method, as previously described [18–22]. Each compound was dissolved in 10% (v/v) dimethyl sulfoxide (DMSO) at 100 mg/mL, and 20 µL aliquots (equivalent to 2 mg/disc) were applied to sterile 6 mm paper discs. A DMSO-only disc (10% v/v) served as the negative control and showed no antimicrobial activity under the test conditions. All discs were UV-sterilized for 20 min prior to use. Test organisms were cultured in tryptic soy broth (TSB), and the turbidity was adjusted to a 0.5 McFarland standard. A 100 µL inoculum was evenly spread onto modified Mueller–Hinton agar (MMHA) plates [19], after which the prepared discs were placed on the inoculated surface. The plates were incubated at 35 ± 2°C for 24 h (bacteria) or 48 h (fungi). All experiments were performed in three independent biological replicates (*n* = 3). Mean zone of inhibition diameters (MZIDs) were measured in millimeters (mm) and reported as the mean ± standard deviation (SD).

**Minimum inhibitory concentration (MIC), minimum biocidal concentration (MBC), minimum biofilm inhibitory concentration (MBIC), and minimum biofilm eradication concentration (MBEC).** MIC, MBC, MBIC, and MBEC values were determined using a resazurin-based microbroth dilution assay combined with spot inoculation, as previously described [18,19,22]. Serial two-fold dilutions of each compound (50–0.098 mg/mL) were prepared in 96-well microtiter

plates. Stock solutions were prepared in DMSO, with the final concentration not exceeding 10% (v/v) in the MIC and MBC assays and maintained at ≤10% (v/v) in the MBIC and MBEC assays, as required by solubility. DMSO-only wells served as negative controls to confirm the absence of antimicrobial or antibiofilm effects. Standardized microbial inocula were added to each well and incubated at 35 ± 2°C for 24 h (bacteria) or 48 h (fungi). After incubation, 25–30 µL of 0.015% (w/v) resazurin was added, followed by an additional 2–4 h of incubation. MIC was defined as the lowest concentration at which the dye remained blue, indicating complete inhibition of metabolic activity. MBC was determined by spot-inoculating aliquots from wells at or above the MIC onto tryptic soy agar (TSA) plates; the lowest concentration showing no colony formation was recorded as the MBC value. MBIC and MBEC values were assessed at the MIC, 2 × MIC, and 4 × MIC, following established protocols [18,19,22]. All assays were performed in triplicate biological replicates ($n = 3$); however, because the replicate outcomes were categorical (growth/no growth) and showed no variability, the results are reported as single consensus values without standard deviation, in accordance with the guidelines for categorical antimicrobial assays. All concentrations are expressed as mg/mL.

## Statistical analysis

All antimicrobial and antibiofilm experiments were performed using three independent biological replicates ($n = 3$). Statistical analysis was applied only to the preliminary antimicrobial activity (zone-of-inhibition data), as these measurements provided continuous numerical values. One-way analysis of variance (ANOVA) was performed using a customized Python script (pandas, matplotlib), followed by Tukey's Honest Significant Difference (HSD) post hoc test using the Statsmodels library. Data distributions were visualized using box and violin plots. Statistical significance was set at $p < 0.05$ [18].

## Molecular docking simulation

Molecular docking simulations were conducted to elucidate the binding orientations and relative affinities of the selected ligands toward the target proteins. Docking analyses were performed for thymol (TM; PubChem CID: 6989), lauryl sulfate (LS; an analog of SLS; PubChem CID: 8778), and phenyl azide (PA; an analog of SA; PubChem CID: 69319) using the Molecular Operating Environment (MOE) software (version 2022.02), in accordance with a previously established and validated protocol [18].

**Selection of target proteins.** Twenty-three druggable target proteins, including key bacterial and fungal proteins involved in biofilm formation, multidrug resistance, and virulence, were selected based on their clinical relevance and prior validation as therapeutic targets. Experimentally determined three-dimensional (3D) structures of proteins were retrieved from the Protein Data Bank (PDB) library [18,23]. All PDB entries used in this study are listed in S1 Table and are cited along with their original structure publications and database references.

Although newer structures with higher nominal resolutions are available for some of the selected targets, the PDB entries used in this study were chosen based on the presence of co-crystallized ligands, well-defined and functionally relevant active-site conformations, and overall structural completeness, particularly in the regions surrounding the binding pocket. These criteria ensured reliable binding-site definition and consistent protein preparation for all targets. Where applicable, newer structures were evaluated; however, the selected entries provided greater methodological consistency for comparative docking and MD analyses, which were central to the study design.

The predicted structures of DabA (MpsA) and DabB (MpsB) were obtained from the AlphaFold Protein Structure Database [24]. AlphaFold models contain per-residue confidence scores (pLDDT) encoded in the B-factor field. To ensure suitability for molecular docking and MD simulations, the structural quality was assessed using a custom Python workflow (BioPython, Matplotlib) [25]. This included the extraction of Cα coordinates, analysis of per-residue pLDDT, calculation of the radius of gyration (Rg) to evaluate global compactness, and computation of the φ/ψ backbone dihedral angles for stereochemical validation of the predicted structures. High-resolution pLDDT profiles and Ramachandran plots were generated using standard confidence thresholds of ≥90 (very high), 70–90 (confident), and 50–70 (low). For all PDB

structures, strict quality filters were applied, retaining only models with a resolution ≤ 2.0 Å and R-free ≤ 0.25 [18,23]. This rigorous selection ensured that all protein structures, both crystallographic and AlphaFold, were reliable and suitable for downstream docking and simulation analyses. Supplementary S1–S4 Figs and Supplementary S2 Table provide complete model-quality results.

**Preparation of protein structures.** Prior to docking, protein structures were prepared by removing crystallographic water molecules and processing them using the MOE QuickPrep module, which included assignment of protonation states at physiological pH (~7.4), correction of structural issues, addition of hydrogen atoms, and energy minimization using default Amber force field parameters [18].

**Preparation of ligand structures.** Ligand structures were prepared using the Protonate 3D module in MOE to assign protonation states at physiological pH (~7.4), followed by energy minimization, during which partial atomic charges were assigned and optimized. The optimized ligands were saved in SDF format for docking [18].

**Ligand modifications.** To address the limitations associated with the ionic nature and structural simplicity of SLS and SA, non-ionic analogs were designed. LS retained the biologically important sulfate group but excluded the sodium ion, thereby enhancing its compatibility for docking studies. PA preserved the azide moiety but introduced an aromatic ring, thereby improving the interaction potential within the protein-binding sites.

**Binding site identification and docking protocol.** Binding sites were defined based on the coordinates of co-crystallized ligands or predicted using the Site Finder tool in MOE. Docking simulations were performed using the Triangle Matcher placement method with London *dG* scoring for initial pose ranking. The top-ranked poses were subsequently refined using an induced-fit protocol to account for receptor flexibility and rescored using the Generalized Born Volume Integral/Weighted Surface Area (GBVI/WSA) *dG* scoring function. For each ligand, 30 poses were generated, and the top five conformations were selected for further refinement. The final docking pose was selected based on the lowest binding free energy and the consistency of key protein–ligand interactions. The selected protein-ligand complexes were energy-minimized and saved in PDB format for subsequent structural and interaction analyses.

**Docking protocol validation.** To ensure the reliability and accuracy of the docking workflow, the docking protocol was validated through redocking experiments [18,23]. The co-crystallized ligands were re-docked into their respective binding sites, and the root-mean-square deviation (RMSD) between predicted and experimental poses was calculated. An RMSD ≤ 2.0 Å was considered indicative of successful reproduction of the native binding mode [22].

Additionally, benchmark ligands with experimentally characterized binding affinities were included to verify the consistency of docking predictions with empirical data. This validation strategy confirms the robustness and reliability of the docking workflow employed in this study.

## Protein–ligand interaction analysis

Protein–ligand interactions were analyzed using a customized Python script inspired by PLIP-style interaction detection, implemented with Biopython, NumPy, and pandas [25–28]. Each docked PDB structure was parsed to separate protein, ligand, and crystallographic water atoms, while common inorganic ions and small buffer components were excluded using a predefined ignore list. Ligands were identified either by user-defined residue names or, when not specified, by automatically selecting the largest non-water hetero-residue in the structure. Hydrogen bonds were detected using a distance-based criterion between polar heavy atoms (N/O) within a cutoff of 3.5 Å, with donor and acceptor roles assigned based on residue- and atom-specific rules. Hydrophobic interactions were defined as contacts between ligand carbon or sulfur atoms and carbon or sulfur atoms of hydrophobic protein residues (Ala, Val, Leu, Ile, Met, Phe, Trp, Pro, Cys, and Tyr) within 4.0 Å, excluding Gly by default. Salt bridges were assigned using distance-based charged-atom criteria, with interactions detected between positively charged Arg or Lys side-chain atoms and ligand oxygen atoms, or between negatively charged Asp or Glu side-chain atoms and ligand nitrogen atoms, within a 4.0 Å cutoff. Water-mediated hydrogen bonds were identified when both ligand–water and protein–water polar contacts were ≤ 3.5 Å. All detected interactions

were compiled into detailed and summary tables and cross-validated against the corresponding two-dimensional (2D) and 3D interaction maps to ensure consistency with the figures presented in the manuscript.

## MD simulation

**Rationale for selection of protein-ligand complexes.** Nine protein-ligand complexes were selected for MD simulations based on their strong binding affinities and relevance as antimicrobial targets. One representative complex was selected from each category of test organisms (Gram-positive bacteria, Gram-negative bacteria, and fungi), prioritizing those with the most negative docking scores and critical ligand–protein interactions [18].

**Ligand preparation and parameterization.** The ligand structures of TM, PA, and LS were prepared and parameterized using CHARMM-compatible force field tools before MD simulations. Atomic partial charges and bonded/non-bonded parameters were assigned using the CHARMM General Force Field (CGenFF), ensuring full compatibility with the CHARMM36m force field employed for protein simulations. The generated ligand topologies were validated for consistency and acceptable CGenFF penalty scores before integration with the protein systems during system assembly.

**Protocol for MD simulation.** MD simulations were performed for all nine protein–ligand complexes containing TM, LS, and PA using GROMACS 2024.2 with the CHARMM36m force field [18]. Each complex was embedded in a truncated octahedral simulation box, solvated with TIP3P water, and ion-adjusted to 0.15 M KCl to mimic physiological conditions. Energy minimization was performed using the steepest-descent algorithm for up to 50,000 steps or until the maximum force fell below 1000 kJ mol$^{-1}$ nm$^{-1}$ [18]. The minimized systems were equilibrated using a two-step protocol comprising 1000 ps under NVT conditions, followed by 1000 ps under NPT conditions (total equilibration time of 2 ns) at 310.15 K, using the Berendsen (velocity-rescaling) thermostat for temperature control and the Parrinello–Rahman barostat for pressure coupling. Equilibration adequacy was verified by monitoring backbone RMSD stabilization and the convergence of temperature, pressure, and system density before initiating the production simulations. All covalent bonds involving hydrogen atoms were constrained using the LINCS algorithm, enabling a 2 fs timestep. Production runs were executed for 300 ns with periodic boundary conditions in all dimensions, and long-range electrostatics were computed using the Particle Mesh Ewald (PME) method with a 1.2 nm cutoff for short-range electrostatic and van der Waals interactions, while long-range electrostatics were treated using PME. The trajectories were saved every 10 ps and analyzed using standard GROMACS tools. Data visualization was performed using GRACE (GRaphing, Advanced Computation and Exploration of data), with subsequent refinement via custom Python scripts to achieve publication-quality figures [18].

**Post-simulation analysis and binding affinity.** Post-simulation analyses were performed to evaluate complex stability and binding affinity. RMSD was used to assess global structural stability, whereas ligand RMSD monitored retention within the active site. Root-mean-square fluctuation (RMSF) was used to identify flexible or dynamic regions of the protein. Rg was calculated to examine protein compactness throughout the simulation, and hydrogen bond analysis quantified the frequency and persistence of protein–ligand hydrogen bonds, reflecting their contribution to complex stabilization. The solvent-accessible surface area (SASA) was computed for both the protein and ligand to characterize solvation behavior and exposure of the binding interfaces [18]. Binding free energies were estimated using the MM/GBSA method implemented in the *gmx_MMPBSA* tool. A 100 ns trajectory window (200–300 ns) comprising 200 evenly spaced, backbone-aligned frames was used after the removal of translational and rotational motions. The calculated energy components included van der Waals energy ($E_{vdW}$), electrostatic energy ($E_{ele}$), polar solvation energy ($G_{polar}$), and non-polar solvation energy ($G_{nonpolar}$). The total binding free energy ($\Delta G_{binding}$) was computed as follows:

$$\Delta G_{binding} = E_{vdW} + E_{ele} + G_{polar} + G_{nonpolar}$$

 

## Determination of dissociation constants ($K_d$) and half maximal inhibitory concentrations ($IC_{50}$)

$K_d$ and $IC_{50}$ values were estimated from the MM/GBSA-derived $\Delta G$ values [18,29]. $K_d$ values were derived using the Gibbs free energy equation:

$$\Delta G = RT \ln K_d$$

where $R = 1.987 \times 10^{-3}$ kcal/mol·K (universal gas constant) and $T = 310$ K (physiological temperature). In principle, $IC_{50}$ can be related to the inhibition constant ($K_i$) using the following equation:

$$IC_{50} = K_i(1 + [S]/K_m)$$

$IC_{50}$ values were estimated under the assumption of competitive inhibition, with the approximation $K_i \approx K_d$ under equilibrium binding conditions, as previously described by Sahakyan et al. [29]. However, owing to the absence of experimental substrate concentration ([S]) and Michaelis–Menten constant ($K_m$) values, $IC_{50}$ was inferred directly from the calculated $K_d$ values and therefore represents preliminary estimates of inhibitory potential rather than exact kinetic parameters.

All thermodynamic and binding calculations ($\Delta G$, $K_d$, and $IC_{50}$) were performed using in-house Python scripts to ensure accuracy and reproducibility in modeling molecular interactions and estimating inhibitory potential from MM/GBSA analyses [18,29]. Although this approach provides a preliminary estimate of the inhibitory potential, experimental validation is necessary to confirm its biological accuracy.

## Principal component analysis (PCA) and free energy landscape (FEL) construction

PCA was conducted to characterize the dominant collective motions of each protein–ligand complex over 300 ns of MD simulations. Trajectories were first corrected for periodic boundary conditions and recentered on the protein using gmx trjconv with the pbc mol and center options, followed by superimposition of all frames onto the Cα atoms of the protein to remove global translational and rotational motions. A covariance matrix of the Cα positional fluctuations was generated and diagonalized using the gmx covar to obtain eigenvalues and eigenvectors, with the first few eigenvectors representing the essential motions of the system. Projections of the trajectories along the first two principal components (PC1 and PC2) were computed using the gmx anaeig, producing projection-versus-time plots and PC1–PC2 conformational scatter maps. FELs were constructed by mapping the conformational population distribution along PC1 and PC2, where the free energy of each state was calculated using the Boltzmann relation:

$$G(PC1, PC2) = -k_B T \ln P(PC1, PC2)$$

where $P(PC1, PC2)$ is the probability of sampling a given conformation, $k_B$ is the Boltzmann constant, and $T$ is the simulation temperature (310.15 K). The visualization of the FEL enabled the identification of the low-energy basins and metastable conformational states of the protein in the presence of each ligand [30,31].

## ADMET analysis

The absorption, distribution, metabolism, excretion, and toxicity (ADMET) properties of TM, SLS, and SA were predicted using ADMETlab 3.0 [18,32] with SMILES strings as the input. Key pharmacokinetic and toxicity parameters, including lipophilicity (LogP), aqueous solubility (LogS), intestinal permeability (Caco-2 cell model), cytochrome P450 (CYP450) enzyme inhibition, and predicted toxicity endpoints, were evaluated. Drug-likeness considerations are discussed separately as supportive physicochemical context and are not treated as a substitute for comprehensive ADMET evaluation. This analysis provides an integrated overview of the predicted pharmacokinetic behavior and safety profiles of the investigated compounds.

## Results and discussion

Antimicrobial resistance (AMR) remains a major global health challenge, diminishing the effectiveness of frontline antibiotics and complicating the management of multidrug-resistant infections [1,2]. In this study, we integrated *in vitro* antimicrobial and antibiofilm assays with *in silico* molecular modeling to evaluate the comparative efficacy of TM, SA, and SLS against MDR and biofilm-forming pathogens. TM demonstrated strong *in vitro* antimicrobial activity and formed stable and biologically meaningful interactions with key microbial targets during long-timescale MD simulations, supporting its potential as a promising antimicrobial scaffold. In contrast, SA and SLS, used here as mechanistic comparators, exhibited apparent limitations, including higher predicted toxicity, weak pharmacokinetic profiles, and inconsistent antimicrobial responses across different microbial classes. However, their inclusion provided essential reference points that helped distinguish target-specific effects from non-specific or physicochemical contributions. Overall, the combined experimental and computational findings provide a coherent mechanistic framework for understanding TM's antimicrobial potential and suggest that LS-like analogs may serve as applicable starting points for future optimization efforts to develop more effective interventions to combat AMR.

### Preliminary antimicrobial activity

The preliminary antimicrobial screening revealed distinct activity profiles for TM, SA, and SLS against Gram-positive bacteria, Gram-negative bacteria, and fungi (Fig 2 and S5 Fig). TM exhibited the strongest and broadest activity, particularly against *C. albicans* (24.8±0.2 mm), *A. niger* (62.0±0.1 mm), *S. epidermidis* (26.3±0.3 mm), and *S. typhimurium* (25.5±0.2 mm). SA also demonstrated notable effects, especially against *C. albicans* (33.5±0.0 mm), *A. niger* (46.0±0.0 mm), *S. aureus* (22.7±0.2 mm), and *S. typhimurium* (29.9±0.1 mm). SLS produced moderate inhibition, with the largest zones recorded against *A. niger* (38.0±0.2 mm), *S. epidermidis* (17.8±0.2 mm), and *P. aeruginosa* (9.8±0.2 mm), whereas the negative control showed no activity (6.0±0.0 mm). TM's potent antifungal effects, particularly against *A. niger*, are consistent with reports of its ability to disrupt fungal membrane integrity and interfere with ergosterol biosynthesis [33]. Its activity against *S. epidermidis* is consistent with that of Xu et al. [22], who demonstrated TM's efficacy against biofilm-forming staphylococci. TM's broad-spectrum activity involves multiple mechanisms, including disruption of cytoplasmic membrane integrity [35], altered membrane permeability [36], quorum-sensing inhibition [37], and interaction with membrane proteins and enzymes essential for cellular function [38]. SA's activity against C. albicans and S. typhimurium is consistent with earlier findings by Lichstein and Soule [39], with its mechanism attributed primarily to cytochrome oxidase inhibition, impairing bacterial respiration [40]; its potency against Gram-negative bacteria also supports observations by Fager and Alben [41]. SLS exhibited moderate activity, likely because of its capacity to solubilize membrane lipids and denature proteins [42]. However, its comparatively lower performance aligns with reports that higher concentrations are required for effective antibacterial action [33]. Its minimal efficacy against P. aeruginosa reflects the organism's intrinsic resistance mechanisms, including low outer membrane permeability [43], efflux pump expression [44], and the presence of antibiotic-inactivating enzymes [45]. Collectively, these findings highlight the markedly superior performance of the natural compound TM compared to the synthetic agents SA and SLS, reinforcing the growing interest in essential-oil-derived antimicrobial scaffolds and suggesting that TM's multimodal mechanisms may reduce the likelihood of resistance development [46].

### MIC, MBC, MBIC, and MBEC

Antimicrobial analyses (S6–S8 Figs) revealed distinct MIC, MBC, MBIC, and MBEC profiles for TM, SA, and SLS against the tested pathogens. TM consistently exhibited low MIC and MBC values (0.10–0.20 mg/mL) against all organisms, demonstrating broad-spectrum bactericidal activity and strong efficacy against MRSA and clinical isolates (S6 Fig). TM also showed potent antibiofilm activity, with MBIC and MBEC values ranging from 0.20–0.39 mg/mL and 0.39–0.78 mg/

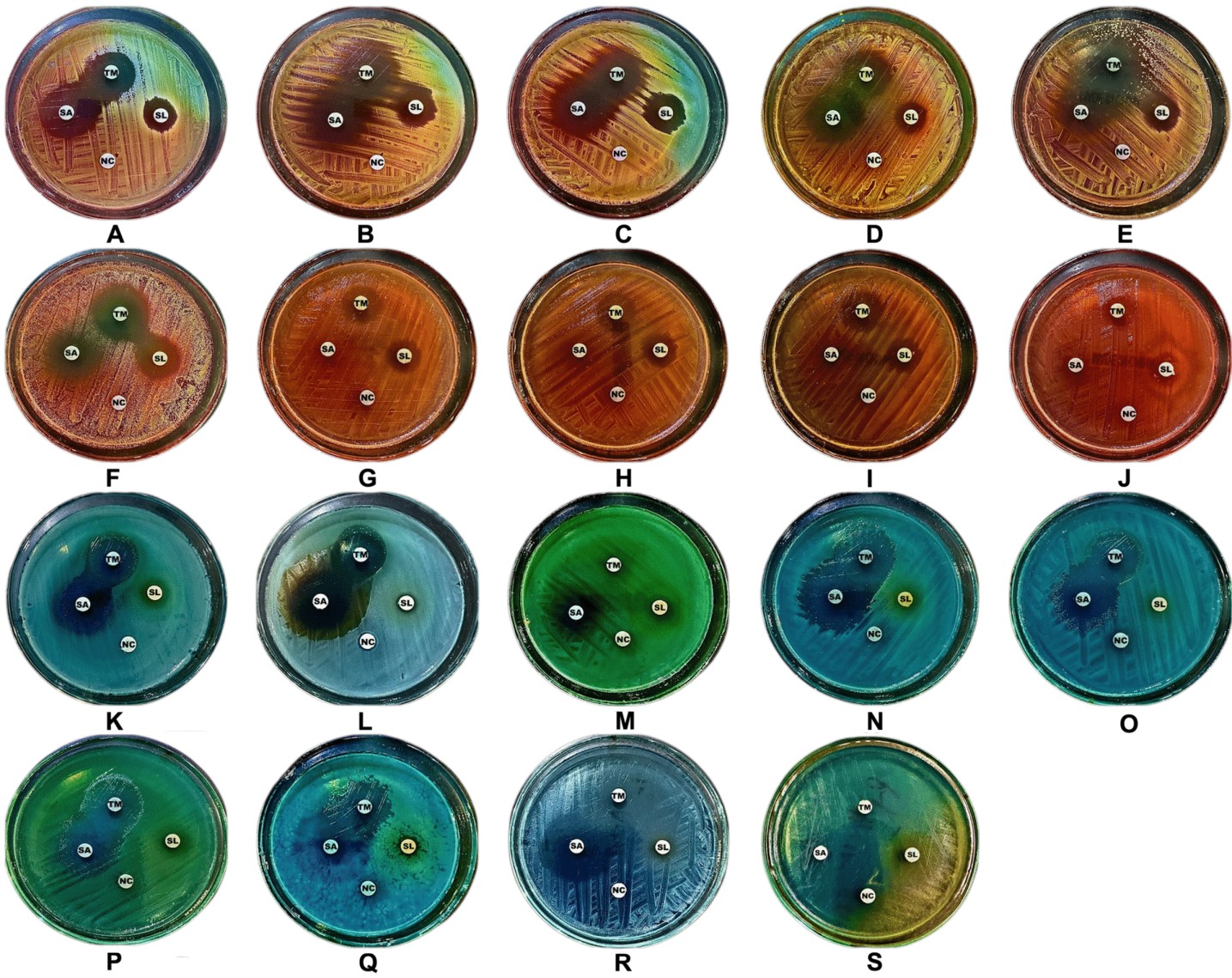

**Fig 2. Preliminary antimicrobial activity of TM, SA, SLS, and NC against the tested pathogens.** Representative zone-of-inhibition images are shown for **(A)** *S. aureus* ATCC 29213, **(B)** *S. aureus*-CI, **(C)** MRSA-1, **(D)** MRSA-2, **(E)** *S. saprophyticus* ATCC 43867, **(F)** *S. epidermidis* ATCC 12228, **(G)** *S. pyogenes*-A ATCC 19615, **(H)** *S. pneumoniae* ATCC 49619, **(I)** *E. faecalis* ATCC 29212, **(J)** *B. cereus* ATCC 10876, **(K)** *E. coli* ATCC 25922, **(L)** *K. pneumoniae* ATCC 27736, **(M)** *P. aeruginosa* ATCC 9027, **(N)** *S. typhimurium* ATCC 13311, **(O)** *S. flexneri* ATCC 12022, **(P)** *P. vulgaris* ATCC 6380, **(Q)** *P. mirabilis* ATCC 29906, **(R)** *C. albicans* ATCC 10231, and **(S)** *A. niger* ATCC 6275. All assays were performed using three independent biological replicates ($n = 3$).

mL, respectively, supporting its potential for treating biofilm-associated infections and aligning with reports documenting its synergistic effects with antibiotics [12,33,34,47]. SA displayed MIC values comparable to TM (0.10–0.78 mg/mL) but required substantially higher concentrations for bactericidal and antibiofilm effects (MBC, MBIC, and MBEC: 6.25–100 mg/ mL), indicating predominantly bacteriostatic activity [39]. SA demonstrated strong antifungal activity (S7 Fig), with low MIC–MBEC values against *C. albicans* and *A. niger*, consistent with its known inhibitory effect on cytochrome oxidase [40]. Simultaneously, its reduced activity against Gram-positive bacteria compared to Gram-negative species may reflect

differences in membrane permeability [43]. SLS showed variable performance, with low MICs for most Gram-positive bacteria (0.10–0.20 mg/mL) but higher MICs for Gram-negative pathogens, especially *P. aeruginosa* (0.78–3.12 mg/mL) and *P. vulgaris* (3.12 mg/mL) (S8 Fig). SLS's MBC, MBIC, and MBEC values were generally high (50–100 mg/mL), particularly against *P. aeruginosa*, highlighting the difficulty of eradicating biofilms in highly resistant strains [45,48]. SLS was notably effective against *A. niger* but showed limited activity against *C. albicans*, likely due to differences in fungal cell wall architecture [49]. Overall, TM exhibited consistently low MIC, MBC, MBIC, and MBEC values across all pathogens, whereas SA and SLS required substantially higher concentrations, particularly for biofilm eradication. These quantitative differences form the foundation for the comparative computational and ADMET analyses presented in the subsequent sections.

## Statistical analysis

The analysis of 57 observations showed that TM had the highest mean antimicrobial activity ($22.22 \pm 10.62$), followed by SA ($18.76 \pm 11.00$), and SLS had the lowest ($12.70 \pm 7.09$). The data were non-normally distributed with unequal variances; however, both ANOVA and Kruskal-Wallis tests confirmed significant differences among the compounds ($F_{(2,168)} = 14.00$, $p < 0.0001$). Tukey's HSD test revealed significant differences between the TM and SLS ($p < 0.001$) groups and the SA and SLS ($p = 0.0031$) groups, but not between the TM and SA ($p = 0.1425$) groups. The Kruskal-Wallis test also confirmed significant differences ($H = 37.76$, $p = 6.32e - 09$). Box and violin plots (S9–S10 Figs) illustrate these trends, proving that TM and SA were more potent than SLS under the tested conditions.

## Molecular dockings analysis

Molecular docking analysis showed that TM, PA, and LS bind stably within the active sites of their respective microbial protein targets, forming energetically favorable complexes supported by hydrogen bonding, hydrophilic, and hydrophobic interactions, and, in selected cases, salt bridge formation (Table 1). Of the 23 docked poses generated for each compound, the nine most favorable complexes—three per ligand, representing Gram-positive bacteria, Gram-negative bacteria, and fungi—were selected for detailed interaction analysis.

TM displayed the strongest binding affinity toward β-carbonic anhydrase (PDB: 5CXK; $\Delta G = -11.2$ kcal/mol), where binding was stabilized by a hydrogen bond with TRP39 and reinforced by a well-defined hydrophobic pocket formed by ALA49, CYS42, LEU52, PRO48, TRP39, and TYR181. TM interaction with FtsZ (PDB: 4DXD; $\Delta G = -11.0$ kcal/mol) was anchored by a hydrogen bond with GLY205 and further supported by hydrophobic contacts involving LEU200, LEU209, VAL203, and VAL297. In fungal sterol 14-α-demethylase (PDB: 5TZ1; $\Delta G = -9.0$ kcal/mol), TM formed hydrogen bonds with HIS377 and SER378 and was stabilized by extensive hydrophobic interactions with LEU376, MET508, PHE233, PHE380, PRO230, and TYR118.

PA exhibited comparatively weaker yet meaningful binding interactions. In β-carbonic anhydrase (PDB: 5CXK; $\Delta G = -9.2$ kcal/mol), PA formed multiple hydrogen bonds with ALA49, ARG64, TRP39, and VAL47, accompanied by hydrophobic contacts involving CYS42, CYS96, and TYR181. The binding of PA to topoisomerase IV (PDB: 4URN; $\Delta G = -7.8$ kcal/mol) involved a hydrogen bond with SER50 and a salt-bridge interaction with ASP76, with additional hydrophobic stabilization provided by MET81. In sterol 14-α-demethylase (PDB: 5TZ1; $\Delta G = -8.8$ kcal/mol), PA binding was supported by hydrogen bonds with ARG381 and HIS468 and hydrophobic contacts involving CYS470, ILE379, LEU376, and PHE463, without detectable π–π stacking interactions in the selected complex.

LS consistently showed strong predicted affinity for all evaluated targets. In peptide deformylase (PDB: 1LMH; $\Delta G = -10.7$ kcal/mol), LS formed multiple hydrogen bonds with GLN65, HIS154, HIS158, and LEU112, supported by hydrophobic interactions with CYS111, LEU105, LEU112, LEU41, and LEU61. The most favorable binding energy was observed for LS in LpxC (PDB: 2VES; $\Delta G = -19.9$ kcal/mol), driven by a dense network of hydrogen bonds with HIS237, HIS78, and THR190, along with extensive hydrophobic contacts involving ALA206, ALA214, ILE197, LEU18, LEU200, MET62, and

**Table 1. Summary of protein–ligand interactions for the top-scoring docked complexes.** Hydrogen-bonding residues, hydrophilic and hydrophobic contacts, salt bridges, and water-mediated hydrogen bonds were identified using a Python-based interaction analysis pipeline inspired by PLIP. π–π stacking interactions are reported where present based on visual inspection of interaction maps. Distances are reported in Å, and ranges indicate multiple contacts involving the same residue. Hydrophobic residues correspond to non-polar ligand–protein contacts (C⋯C/S interactions), whereas hydrophilic residues represent hydrogen bond donors/acceptors or charged side chains contributing to polar interactions. n denotes the number of individual hydrogen bonds or salt-bridge contacts contributed by a given residue.

| # | Docked Complex (Ligand-Protein) | Hydrogen-bonding residues (distance, Å) | π–π stacking residues (distance, Å) | Residues involved in hydrophilic interactions | Residues involved in hydrophobic interactions | Salt-bridge residues (distance, Å) | Water-mediated H-bonds (protein residues) |
|---|---|---|---|---|---|---|---|
| 1 | 5CXK-TM | TRP39 (3.43 Å) | — | TRP39 | ALA49, CYS42, LEU52, PRO48, TRP39, TYR181 | — | — |
| 2 | 4DXD-TM | GLY205 (3.35 Å) | — | GLY205 | LEU200, LEU209, VAL203, VAL297 | — | — |
| 3 | 5TZ1-TM | HIS377 (3.12 Å), SER378 (3.11 Å) | — | HIS377, SER378 | LEU376, MET508, PHE233, PHE380, PRO230, TYR118 | — | — |
| 4 | 5CXK_PA | ALA49 (2.98 Å), ARG64 (2.95–3.17 Å; n = 2), TRP39 (3.07–3.27 Å; n = 2), VAL47 (3.01–3.25 Å; n = 2) | — | ALA49, ARG64, TRP39, VAL47 | CYS42, CYS96, TYR181 | — | — |
| 5 | 4URN_PA | SER50 (3.35 Å) | — | ASP76, SER50 | MET81 | ASP76 (3.72 Å) | — |
| 6 | 5TZ1_PA | ARG381 (3.26–3.37 Å; n = 2), HIS468 (3.28–3.31 Å; n = 2) | — | ARG381, HIS468 | CYS470, ILE379, LEU376, PHE463 | — | — |
| 7 | 1LMH_LS | GLN65 (3.01–3.01 Å; n = 2), HIS154 (2.85–3.30 Å; n = 2), HIS158 (2.95 Å), LEU112 (2.76 Å) | — | GLN65, HIS154, HIS158, LEU112 | CYS111, LEU105, LEU112, LEU41, LEU61 | — | — |
| 8 | 2VES_LS | HIS237 (2.92–3.42 Å; n = 2), HIS78 (2.68–3.50 Å; n = 2), THR190 (2.59–3.08 Å; n = 2) | — | HIS237, HIS78, THR190 | ALA206, ALA214, ILE197, LEU18, LEU200, MET62, PHE191 | — | — |
| 9 | 5JPF_LS | ARG386 (2.73 Å), HIS231 (2.86 Å), HIS290 (2.94–3.25 Å; n = 2), HIS413 (2.95 Å) | — | ARG386, HIS231, HIS290, HIS413 | TRP371, TYR299 | ARG386 (2.73–3.71 Å; n = 2) | — |

PHE191. LS also strongly bound to serine/threonine phosphatase Z1 (PDB: 5JPF; $\Delta G$ = −13.4 kcal/mol), forming hydrogen bonds with ARG386, HIS231, HIS290, and HIS413, complemented by hydrophobic interactions with TRP371 and TYR299 and a stabilizing salt-bridge interaction involving ARG386.

Overall, these results reveal clear ligand- and target-specific interaction patterns across bacterial and fungal proteins. The comparative docking trends are illustrated in Fig 3, and Figs 4–6 present detailed two- and three-dimensional interaction maps. All nine selected complexes were subsequently subjected to 300 ns MD simulations to evaluate their structural stability, interaction persistence, and conformational behavior under simulated physiological conditions, providing further insight into their potential as antimicrobial candidates.

## MD simulation and binding affinity analysis

Three-hundred-nanosecond MD simulations were performed to evaluate the structural stability, flexibility, and interaction energetics of the nine selected protein–ligand complexes under physiologically relevant conditions. These analyses included RMSD, RMSF, Rg, SASA, hydrogen-bond profiles, and MM/GBSA binding free energies, providing a comprehensive view of ligand-induced stabilization and dynamic behavior across the complexes.

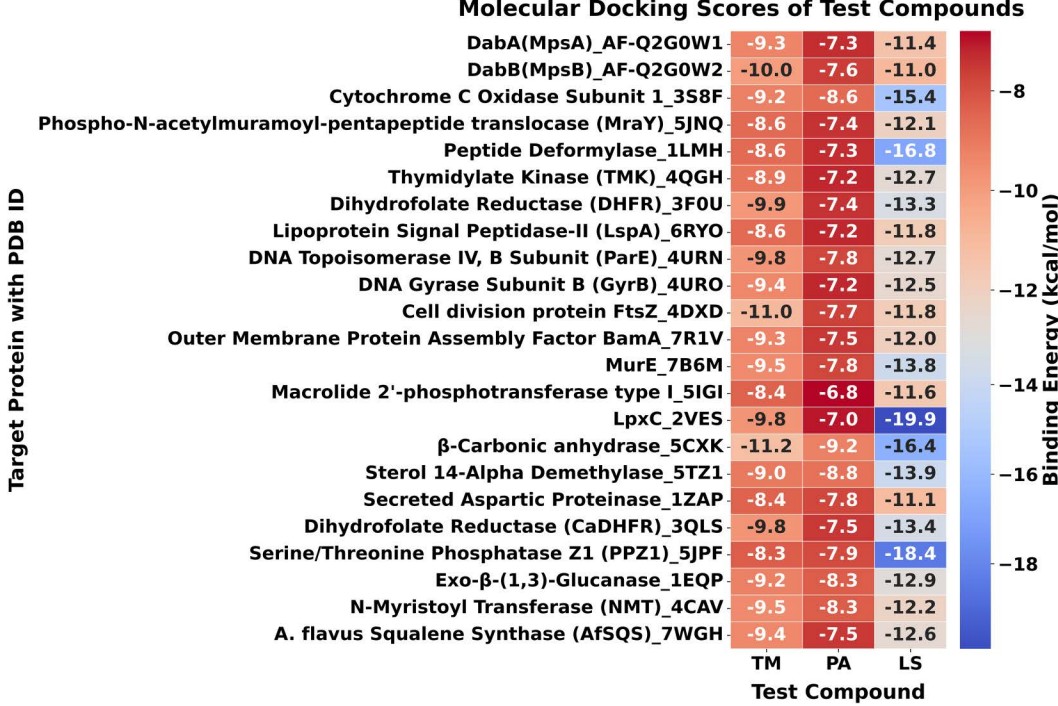

**Fig 3. Heatmap of molecular docking scores for TM, PA, and LS against all target proteins.** Lower docking scores indicate stronger predicted binding affinity. The heatmap summarizes the comparative binding strengths of each compound across the full protein panel.

**Overall MD equilibration, flexibility, and hydrogen bond statistics.** All simulated systems displayed clear equilibration, with the backbone RMSD stabilizing within 30–50 ns and remaining consistent thereafter, indicating stable conformational sampling across the 300 ns trajectories. The RMSDs showed slightly larger fluctuations in 4DXD–TM, 4URN–PA, and 5JPF–LS, reflecting ligand mobility rather than protein instability, as none of the systems showed long-term drift. The RMSF profiles revealed the highest flexibility in the terminal and loop regions. The active-site residues exhibited reduced fluctuations, particularly in 4URN–PA, 5TZ1–PA, and all LS-bound complexes, indicating ligand-induced stabilization. The hydrogen-bond trends showed ligand-specific behavior: the TM complexes formed a few, mostly transient H-bonds (0–1); the PA complexes exhibited moderate but variable H-bonding (1–4); and the LS complexes, especially 2VES–LS, maintained the highest number of stable H-bonds (2–5). The Rg and SASA values remained stable across all trajectories, further confirming global structural compaction and good convergence. Collectively, these results demonstrate the overall stability of all complexes and highlight the apparent differences in ligand-induced dynamic behavior.

**TM complexes.** TM-bound systems (5CXK–TM, 4DXD–TM, and 5TZ1–TM; S11–S13 Figs) exhibited overall stable conformational behavior during the 300 ns MD simulations. RMSD analyses indicated that all three complexes reached equilibrium and maintained stable trajectories, with the ligands remaining bound within the active site throughout the simulation. The 4DXD–TM complex showed moderately higher RMSD fluctuations than 5CXK–TM and 5TZ1–TM, particularly during the early and intermediate phases of the simulation; however, these fluctuations did not result in ligand dissociation or major structural disruption. RMSF profiles demonstrated reduced backbone flexibility in the ligand-binding regions relative to the unbound protein, indicating local stabilization upon TM binding in all systems. Rg analysis confirmed that the protein structures remained compact over time, whereas SASA profiles showed only minor variations, suggesting limited changes in overall solvent exposure following ligand association.

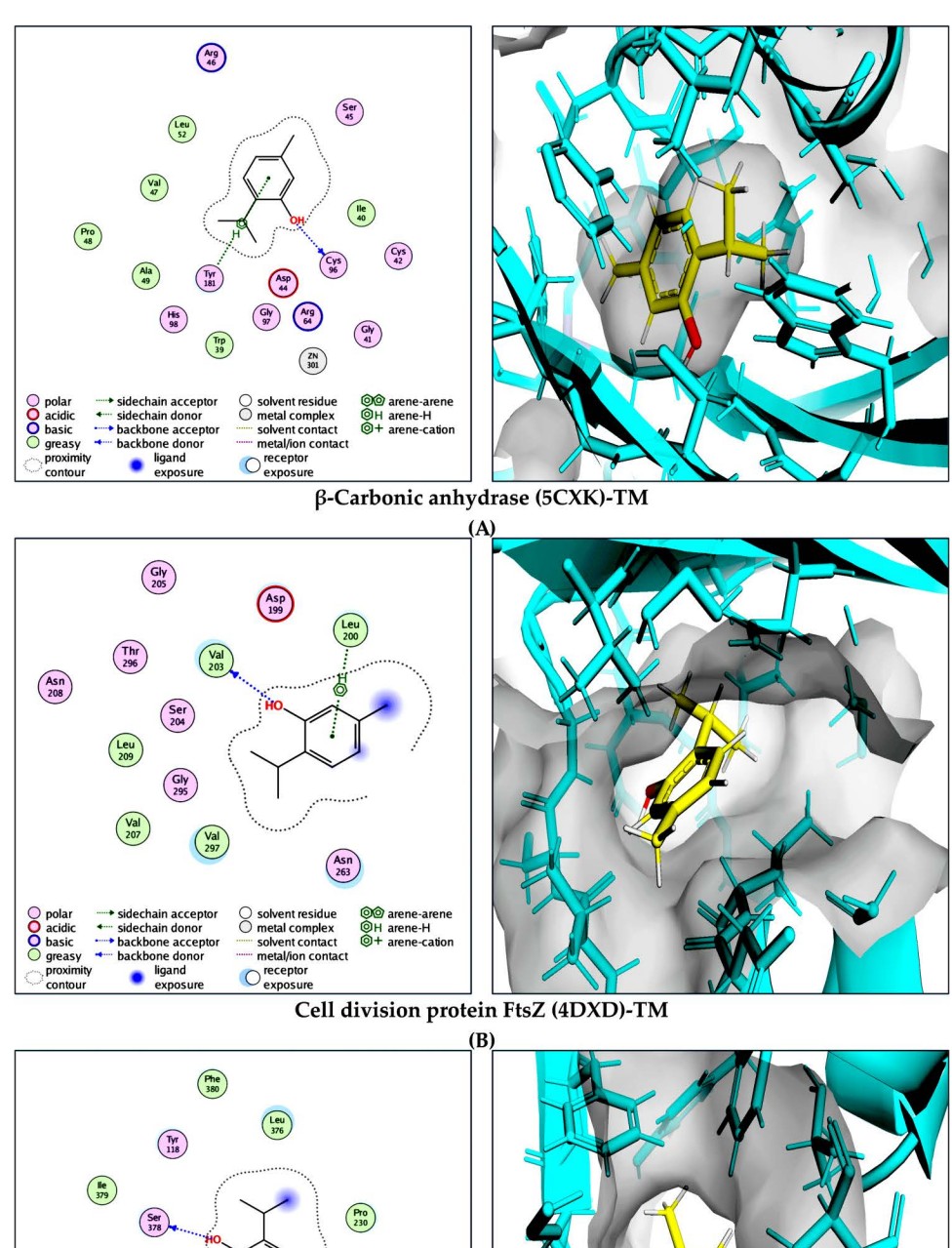

**β-Carbonic anhydrase (5CXK)-TM**

**(A)**

**Cell division protein FtsZ (4DXD)-TM**

**(B)**

**Sterol 14-Alpha Demethylase (5TZ1)-TM**

**(C)**

**Fig 4. Molecular interaction analysis of TM with selected microbial target proteins.** Combined 2D and 3D binding interaction representations illustrating the binding mode, key stabilizing interactions, and active-site accommodation of TM within: **(A)** β-carbonic anhydrase (PDB: 5CXK), **(B)** cell

division protein FtsZ (PDB: 4DXD), and **(C)** sterol 14-α-demethylase (PDB: 5TZ1). The 2D interaction maps highlight hydrogen bonding, hydrophobic contacts, and aromatic interactions, whereas the corresponding 3D views depict TM within the respective binding pockets, demonstrating stable, target-specific binding conformations.

Hydrogen bond analysis revealed predominantly transient interactions, with an average of 0–1 hydrogen bond observed during the simulations, indicating that TM binding is not primarily stabilized by persistent hydrogen bonding. In contrast, hydrophobic interaction distances remained stable throughout the trajectories, highlighting hydrophobic contacts as the dominant stabilizing force in TM-bound complexes. MM/GBSA calculations confirmed favorable binding energies for all TM complexes, driven mainly by van der Waals interactions with additional electrostatic contributions. Among the three systems, 4DXD–TM and 5TZ1–TM displayed more favorable binding free energies than 5CXK–TM, consistent with their stronger predicted binding affinities. Overall, TM formed stable protein–ligand complexes characterized by compact conformations, limited residue-level flexibility, and binding dominated by hydrophobic interactions with occasional hydrogen bond support.

**PA complexes.** PA-bound complexes (5CXK–PA, 4URN–PA, and 5TZ1–PA; S14–S16 Figs) exhibited target-dependent dynamic behavior over 300 ns of simulations. RMSD profiles indicated that 5CXK–PA and 5TZ1–PA maintained stable binding trajectories after initial equilibration, whereas 4URN–PA exhibited comparatively larger fluctuations in the complex RMSD. RMSF trends were broadly consistent with these observations, with lower flexibility in the bound-state profiles for 5CXK–PA and 5TZ1–PA relative to the more dynamic behavior observed in 4URN–PA. Across all systems, Rg and SASA remained largely stable, indicating no major global unfolding events.

Hydrogen-bond analysis revealed clear differences in interaction persistence: 5TZ1–PA sustained the highest hydrogen-bond occupancy (typically ~2–4), 5CXK–PA maintained moderate hydrogen bonding (~1–3), and 4URN–PA showed predominantly transient hydrogen bonds (mostly ~0–1). Importantly, MM/GBSA binding free energy and $K_d$ estimates identified 4URN–PA as the least favorable complex, with near-zero $\Delta G$(binding) and the largest $K_d$, indicating a weak predicted affinity. In contrast, 5TZ1–PA showed the most favorable PA binding, followed by 5CXK–PA. Overall, PA exhibited context-dependent binding, with strong stabilization in 5TZ1–PA, moderate stability in 5CXK–PA, and weak binding characteristics in 4URN–PA.

**LS complexes.** LS-bound complexes (1LMH–LS, 2VES–LS, and 5JPF–LS; S17–S19 Figs) exhibited stable structural behavior throughout the 300 ns MD simulations. RMSD analyses showed that all three systems reached equilibrium early and maintained relatively stable trajectories, with 2VES–LS displaying the lowest overall RMSD values, indicating the highest conformational stability among the LS-bound systems. Ligand RMSD profiles further confirmed consistent ligand retention within the binding pocket across all complexes. RMSF profiles revealed generally low residue-level fluctuations, suggesting limited flexibility within the binding regions upon LS binding. Rg and SASA analyses demonstrated compact protein conformations with minimal variation over time, indicating that ligand association did not induce large-scale structural rearrangements.

Hydrogen bond analysis highlighted notable differences among the systems: 2VES–LS maintained a higher and more persistent number of hydrogen bonds over the simulation, whereas 1LMH–LS showed moderate hydrogen bond stability and 5JPF–LS exhibited more transient hydrogen-bonding behavior. Hydrophobic interaction distances remained stable across all LS-bound complexes, supporting a consistent contribution of non-polar contacts to binding stability. MM/GBSA calculations indicated favorable binding free energies for all LS complexes, with 2VES–LS exhibiting the most favorable $\Delta G$(binding), followed by 1LMH–LS and 5JPF–LS. Consistent with these results, $K_d$ estimates placed 2VES–LS in the picomolar range, identifying it as the strongest LS-bound complex, while 1LMH–LS showed nanomolar affinity and 5JPF–LS comparatively weaker binding. Overall, LS binding resulted in stable protein–ligand complexes, with binding strength primarily governed by a combination of persistent hydrogen bonding and hydrophobic interactions.

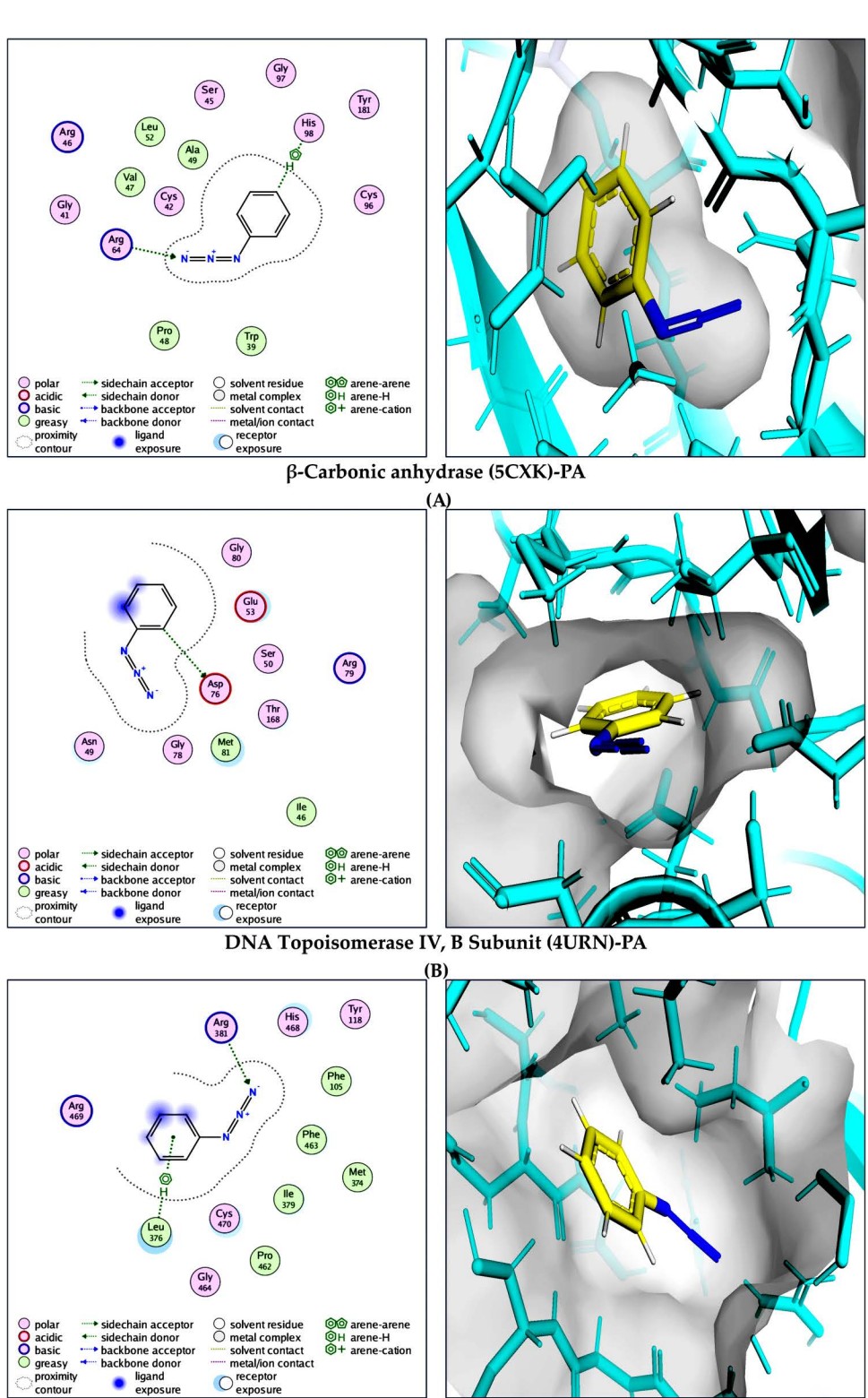

**β-Carbonic anhydrase (5CXK)-PA**

**(A)**

**DNA Topoisomerase IV, B Subunit (4URN)-PA**

**(B)**

**Sterol 14-Alpha Demethylase (5TZ1)-PA**

**(C)**

**Fig 5. Molecular interaction analysis of PA with selected microbial target proteins.** Combined 2D and 3D binding interaction representations illustrating the binding mode, key stabilizing interactions, and active-site accommodation of PA within: **(A)** β-carbonic anhydrase (PDB: 5CXK), **(B)** DNA

topoisomerase IV, B subunit (PDB: 4URN), and **(C)** sterol 14-α-demethylase (PDB: 5TZ1). The 2D interaction maps highlight hydrogen bonding, hydrophobic contacts, and aromatic interactions, whereas the corresponding 3D views show PA positioned within the respective binding pockets, demonstrating stable, target-specific binding conformations.

**Comparative binding affinity and structure-activity relationships.** A comparative evaluation of the predicted binding affinities revealed clear differences among the studied protein–ligand systems. Based on the $K_d$ and inferred $IC_{50}$ values, the LS-bound complex 2VES–LS ($K_d = 1.7 \times 10^{-11}$ M) exhibited the strongest predicted affinity across all evaluated systems, followed by the TM-bound complex 4DXD–TM ($K_d = 3.2 \times 10^{-10}$ M). Additional high-affinity interactions were observed for the LS-bound 1LMH and TM- or PA-bound 5TZ1 complexes, which consistently demonstrated favorable $\Delta G$(binding) values and low dissociation constants. In contrast, the 5CXK complexes, 5JPF–LS, and 4URN–PA exhibited comparatively weaker predicted affinities, in agreement with their less favorable MM/GBSA binding free energies.

Across ligand classes, LS displayed the strongest and most consistent binding behavior across different protein targets, reflecting its ability to establish stable noncovalent interaction networks. TM exhibited target-dependent high-affinity interactions, performing particularly well in selected systems, whereas PA showed more variable stabilization effects that were highly context-dependent. Importantly, all $IC_{50}$ values were derived for comparative ranking purposes only and should not be interpreted as absolute inhibitory potencies. Given the indirect estimation of $IC_{50}$ from $K_d$ and the nonlinear relationship between $\Delta G$, $K_d$, and biological inhibition, these results represent relative indicators of binding strength rather than quantitative measures of activity.

Fig 7 summarizes these trends by presenting a comparative heatmap of MM/GBSA-derived $\Delta G$(binding) values alongside the predicted $K_d$ estimates for all protein–ligand complexes. Although the computational results provide meaningful structure–activity insights and a relative ranking of ligand performance, experimental validation is required to confirm the predicted inhibitory potential of the identified lead systems.

## Essential dynamics and free energy landscape analysis using PCA

PCA, combined with FEL mapping, were used to investigate the dominant motions, conformational stability, and energy minima of all protein–ligand complexes during the 300 ns MD simulations. The projection of trajectories onto PC1 and PC2 captured large-scale collective motions, whereas the FEL plots revealed the energetic distribution of sampled conformations, with deeper, more compact minima indicating higher structural stability.

**PCA–FEL analysis of TM complexes.** *5CXK–TM:* The 5CXK–TM complex (S20 Fig) exhibited the largest conformational fluctuations among the TM systems, with PC1 mean ± SD = −0.00 ± 7.39 and PC2 = −0.00 ± 3.35. A major conformational transition occurred at approximately 140 ns, after which the motion stabilized. The FEL displayed multiple shallow to moderately deep energy minima (~0–15 kJ/mol), indicating broad conformational exploration. The scatter plot shows a wide distribution, consistent with heterogeneous sampling of the structural states.

*4DXD–TM:* 4DXD–TM (S21 Fig) showed the most restricted motions (PC1 = −0.00 ± 1.73 and PC2 = 0.00 ± 0.98), which were characterized by tightly clustered essential dynamics. The FEL consisted of two compact, deep basins (~0–8 kJ/mol), indicating a highly stable conformational ensemble. The scatter plot displays dense clustering, confirming minimal transitions between the states.

*5TZ1–TM:* 5TZ1–TM (S22 Fig) demonstrated intermediate behavior, with PC1 = −0.00 ± 1.24 and PC2 = 0.00 ± 0.96. Moderate oscillations were observed during the simulation. The FEL contained several shallow minima (~0–7 kJ/mol), indicating flexible but stable conformational states. The scatter plots showed moderate dispersion between the extremes of 5CXK–TM and 4DXD–TM. Overall, 5CXK–TM underwent broad conformational exploration, 4DXD–TM achieved the strongest stabilization, and 5TZ1–TM represented an intermediate state, highlighting the protein-specific effects of TM binding on essential dynamics and free-energy landscapes.

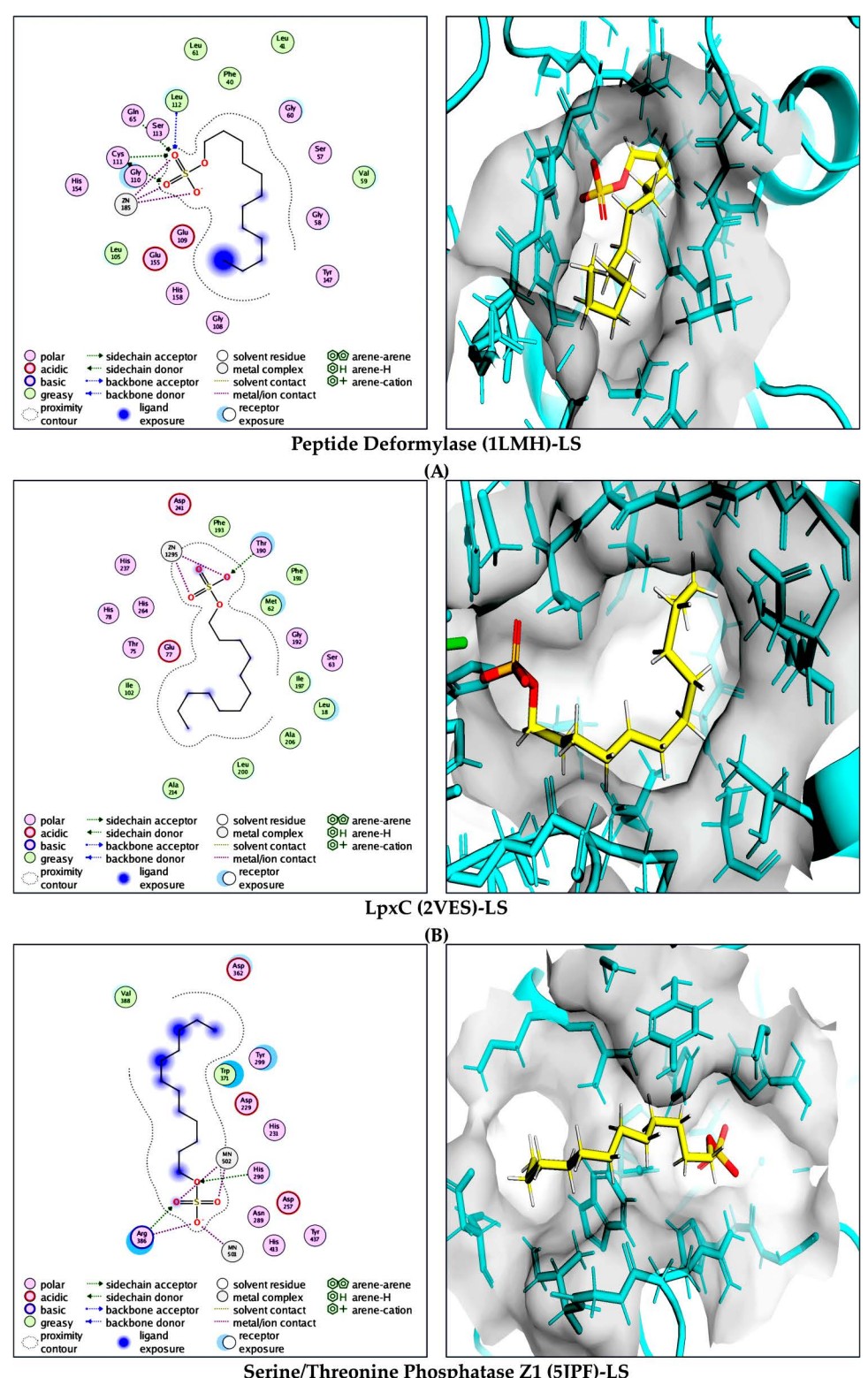

**Fig 6. Molecular interaction analysis of LS with selected microbial target proteins.** Combined 2D and 3D binding interaction representations illustrating the binding mode, key stabilizing interactions, and active-site accommodation of LS within: **(A)** peptide deformylase (PDB: 1LMH), **(B)** lipid A

deacetylase LpxC (PDB: 2VES), and **(C)** serine/threonine phosphatase Z1 (PDB: 5JPF). The 2D interaction maps highlight hydrogen bonding, hydrophobic contacts, and electrostatic interactions, whereas the corresponding 3D views depict LS within the respective catalytic pockets, demonstrating stable, target-specific binding conformations.

### Binding Affinity Analysis ($\Delta G$ and $K_d$) of Protein-Ligand Complexes

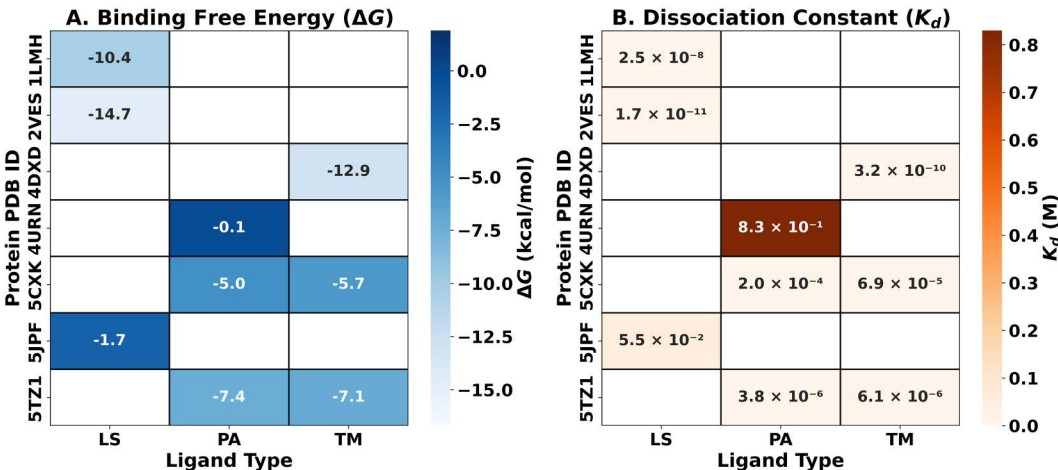

**Fig 7. Heatmap comparing MM/GBSA binding free energies ($\Delta G$) and estimated dissociation constants ($K_d$) for selected protein–ligand docked complexes.** Lower (more negative) $\Delta G$ values and smaller $K_d$ values indicated stronger predicted binding affinities across the evaluated complexes.

**PCA–FEL analysis of PA complexes.** *4URN–PA:* 4URN–PA (S23 Fig) showed very restricted dynamics (PC1 mean±SD = −0.00±0.43; PC2 = −0.00±0.36), with nearly flat fluctuation profiles over 300 ns. The FEL revealed a single, compact, and shallow basin (~0–3 kJ/mol), indicating strong conformational stability. The scatter plots demonstrated narrow clustering.

*5CXK–PA:* 5CXK–PA (S24 Fig) exhibited the broadest essential motions among the PA systems (PC1 = 0.00±4.91 and PC2 = −0.00±1.66). Large-amplitude transitions occurred within the first 50 ns. Its FEL displayed two distinct, moderately deep basins (~0–12 kJ/mol), reflecting transitions between metastable states. The scatter plot shows wide dispersion along PC1.

*5TZ1–PA:* 5TZ1–PA (S25 Fig) demonstrated intermediate flexibility (PC1 = −0.00±1.48 and PC2 = 0.00±1.06). Several shallow FEL minima (~0–8 kJ/mol) were observed, indicating a balance between stability and mobility. Scatter plots revealed moderate conformational sampling, which was narrower than that of 5CXK–PA but broader than that of 4URN–PA. Scree plots further confirmed these trends: 5CXK–PA dynamics were dominated by large-scale collective motions (PC1 accounting for the majority of the variance), 4URN–PA distributed variance across several low-amplitude modes, and 5TZ1–PA reflected intermediate contributions. Overall, 4URN–PA was the most stable, exhibiting minimal essential motions. In contrast, 5CXK–PA sampled the largest conformational subspace, exhibiting distinct metastable states. 5TZ1–PA balanced flexibility and stability, highlighting protein-specific responses upon PA binding.

**PCA–FEL analysis of LS complexes.** *1LMH–LS:* 1LMH–LS (S26 Fig) exhibited moderate conformational fluctuations (PC1 mean±SD = −0.00±1.48, and PC2 = 0.00±1.06). The FEL showed multiple shallow minima (~0–6 kJ/mol), suggesting flexible yet stable dynamics. The scatter plots show dispersed sampling across PC1–PC2.

*2VES–LS:* 2VES–LS (S27 Fig) displayed the most restricted motion of all the complexes (PC1 = 0.00±0.67 and PC2 = −0.00±0.47). The FEL revealed two compact, low-energy basins (~0–5 kJ/mol), confirming exceptional conformational stabilization by LS. The scatter plots show tight clustering into two dominant states, indicating minimal conformational drift.

*5JPF–LS:* 5JPF–LS (S28 Fig) showed intermediate fluctuations (PC1 = 0.00 ± 0.85; PC2 = −0.00 ± 0.67). The FEL mapped several moderately deep minima (~0–7 kJ/mol), indicating a broader conformational landscape than 2VES–LS but more restricted than 1LMH–LS. The scatter distributions confirmed intermediate conformational sampling. Overall, 2VES–LS was the most stable system, exhibiting restricted conformational dynamics, whereas 1LMH–LS displayed flexible yet controlled motions. 5JPF–LS exhibited intermediate behavior, with a broader sampling of conformational states, reflecting the protein-specific effects of LS binding.

Across all systems, FEL and PCA analyses consistently identified 2VES–LS as the most stable complex, characterized by compact, low-energy minima, and minimal essential motions. The TM and PA complexes exhibited protein-dependent behaviors, with 4DXD–TM and 4URN–PA forming exceptionally stable conformational basins. In contrast, 5CXK–TM and 5CXK–PA sampled the broadest conformational landscapes, indicating a higher dynamic mobility. These findings align well with the RMSD, RMSF, and binding energy trends, confirming the ligand-specific stabilization of protein structures.

Additional post-MD descriptors, such as center-of-mass tracking and per-frame binding energy decomposition, were not included because the combined MM/GBSA binding free energies, hydrogen-bond occupancy, RMSD/RMSF, PCA, and free-energy landscape analyses already provided a comprehensive and non-redundant assessment of binding stability and conformational behavior.

## ADMET analysis

All pharmacokinetic and toxicity parameters reported in this study were derived exclusively from *in silico* predictions using ADMETlab 3.0 with SMILES strings as input, and no *in vitro* or *in vivo* toxicity experiments were performed. Accordingly, all toxicity values should be interpreted strictly as computational estimates rather than experimentally validated measurements. ADMET profiling revealed distinct pharmacokinetic and safety characteristics among TM, SA, and SLS. TM exhibited moderate lipophilicity (LogP = 3.16), acceptable predicted solubility, and favorable synthetic accessibility. However, predicted cytochrome P450 inhibition and high plasma protein binding indicate a potential risk of drug–drug interactions. Additional predictions suggested moderate cellular permeability, limited oral absorption, and localized irritation alerts (ocular, dermal, and respiratory), supporting the feasibility of topical or localized therapeutic applications (S29 Fig). SA showed unfavorable pharmacokinetic characteristics, including poor predicted absorption, multiple structural alerts, and high predicted toxicity across hepatic, renal, neurological, and genotoxic endpoints, consistent with its established use as a laboratory biocide rather than a therapeutic agent (S30 Fig). SLS displayed moderate physicochemical parameters but exhibited poor predicted bioavailability, very high plasma protein binding (97.96%), and inhibition of multiple transporters and CYP isoenzymes. Predicted toxicities included hepatic injury, irritation, and environmental hazards, aligning with its known applications in topical or industrial formulations rather than systemic therapy (S31 Fig). SA demonstrated potent antifungal activity but inconsistent antibacterial activity, likely due to its mechanism of action as a respiration inhibitor. In contrast, SLS showed membrane-disruptive activity, consistent with its established non-systemic applications [49].

A comparative assessment integrating experimental antimicrobial data and computational analyses indicated that TM consistently exhibited superior antimicrobial performance among the tested compounds. This observation is supported by its lower MIC values (0.10–0.20 mg/mL) and stable predicted binding interactions with key microbial targets, including FtsZ ($K_d = 3.2 \times 10^{-10}$ M) and sterol 14-α-demethylase ($K_d = 6.1 \times 10^{-6}$ M), in agreement with previous reports [10]. SA demonstrated notable antifungal activity but variable antibacterial efficacy, likely reflecting its mechanism of action as a respiration inhibitor rather than a target-specific antimicrobial agent. In contrast, SLS primarily exhibits membrane-disruptive effects and fails to meet key pharmacological and toxicity criteria, which is consistent with its established non-systemic and industrial applications [49].

## Clinical translation and formulation prospects for TM

Given TM's lipophilicity and moderate oral bioavailability [10], clinically relevant applications are more feasible through topical and localized delivery systems. Recent studies have demonstrated that nanoemulsions [50,51], liposomal carriers [52], polymeric nanoparticles [13], hydrogels [53], and bioadhesive films [54] enhance TM's solubility, stability, and penetration into infected tissues while minimizing irritation. These platforms are beneficial for treating skin, wound, and device-associated biofilm infections. TM has also shown synergistic activity with frontline antibiotics, including ciprofloxacin, vancomycin, linezolid, and cefazolin [12,55], suggesting that combination therapies may reduce effective doses and help overcome resistance. Together, these formulation strategies and synergistic regimens provide feasible translational pathways to advance TM for therapeutic development.

## Limitations of the study

Although this study integrated *in vitro* antimicrobial and antibiofilm assays with comprehensive *in silico* analyses, several limitations must be acknowledged. SA and SLS were included primarily as comparative controls because of their known toxicity, and their observed activity should not be interpreted as evidence of therapeutic potential. The exceptionally high docking-derived affinities predicted for LS analogs may partially reflect computational bias arising from rigid receptor assumptions, scoring function limitations, and restricted conformational sampling; thus, caution should be exercised in their interpretation and experimental validation. $IC_{50}$ estimations were inferred from $\Delta G$ values under the approximation that $K_i \approx K_d$ and are intended solely for comparative ranking, rather than as absolute measures of inhibitory potency. In addition, although PCA and FEL analyses were performed to characterize large-scale conformational motions, additional convergence diagnostics, such as MD trajectory clustering, were not conducted and should be incorporated in future studies to further enhance robustness. Despite these constraints, this study provides meaningful comparative insights, supports TM as a viable repurposable antimicrobial candidate, and identifies LS-based analogs as promising scaffolds for rational optimization.

## Conclusion

This study presents an integrated experimental and computational evaluation of thymol (TM), sodium azide (SA), and sodium lauryl sulfate (SLS), combining *in vitro* antimicrobial and antibiofilm assays with molecular docking, 300 ns MD simulations, MM/GBSA binding free-energy calculations, and PCA/free-energy landscape analyses. Among the tested compounds, TM demonstrated superior antimicrobial performance, as reflected by low MIC/MBC values, strong antibiofilm activity, and stable target-specific interactions with essential microbial proteins, particularly FtsZ and sterol 14-α-demethylase. Computational analyses revealed that TM-mediated stabilization was primarily driven by hydrophobic interactions supplemented by transient hydrogen bonding, supporting a robust, mechanistically coherent mode of action.

In contrast, although SA exhibits notable antifungal activity, its poor predicted pharmacokinetic properties and high toxicity substantially limit its therapeutic potential. SLS showed variable antimicrobial activity and unfavorable ADMET properties; however, its neutral analog, LS, showed strong predicted binding to targets such as LpxC. These interactions, while partially influenced by computational overestimation, highlight LS-like scaffolds as applicable starting points for rational structural optimization, rather than as immediate lead candidates. Collectively, TM has emerged as the most promising compound based on its combined experimental efficacy, mechanistic stability, and comparatively favorable predicted safety profile.

Despite these encouraging findings, this study is limited by its reliance on *in silico* ADMET predictions and the absence of *in vivo* validation. Future investigations should therefore prioritize *in vivo* efficacy and toxicity assessments of TM, explore synergistic combinations with existing antibiotics, and pursue systematic medicinal-chemistry optimization of LS-derived scaffolds. Overall, TM emerges as the most balanced repurposable antimicrobial candidate, whereas LS-based analogs represent computationally prioritized scaffolds that warrant further experimental validation to address the growing challenge of antimicrobial resistance.

## Supporting information

**S1 Fig. pLDDT confidence profile of DabA (MpsA) predicted by AlphaFold.** Per-residue pLDDT scores are plotted across the DabA sequence, with standard confidence ranges indicated (very high ≥90; high 70–90; low 50–70; very low<50). The profile shows predominantly high-confidence regions, supporting a well-folded and structurally reliable model.
(TIF)

**S2 Fig. Ramachandran plot of the AlphaFold-predicted DabA (MpsA) structure.** Backbone φ/ψ dihedral angles for all non-glycine and non-proline residues are displayed. Most residues fall within favored α-helical and β-sheet regions, indicating good stereochemical geometry and supporting the structural quality of the model.
(TIF)

**S3 Fig. pLDDT confidence profile of DabB (MpsB) predicted by AlphaFold.** Residue-wise pLDDT values are shown with confidence zones marked. DabB exhibits consistently high-confidence predictions across structured domains, with moderate-confidence values limited to expected flexible or loop regions.
(TIF)

**S4 Fig. Ramachandran plot of the AlphaFold-predicted DabB (MpsB) structure.** The φ/ψ dihedral angle distribution indicates that most residues occupy favored secondary-structure regions, demonstrating good stereochemical quality. Only a small number of residues fall in outlier regions, primarily within flexible loop segments.
(TIF)

**S5 Fig. Heatmap of preliminary antimicrobial activity of TM, SA, and SLS.** Mean zone of inhibition diameters (MZIDs, mm) are shown for each test organism. All assays were performed in three independent biological replicates ($n = 3$).
(TIF)

**S6 Fig. Heatmap of antimicrobial potency of TM.** Heatmap showing MIC, MBC, MBIC, and MBEC values (mg/mL) of TM against the tested pathogens. Lower values indicate higher antimicrobial potency. All assays were performed in three independent biological replicates ($n = 3$); consensus values are reported.
(TIF)

**S7 Fig. Heatmap of antimicrobial potency of SA.** Heatmap showing MIC, MBC, MBIC, and MBEC values (mg/mL) of SA against the tested pathogens. Lower values indicate higher antimicrobial potency. All assays were performed in three independent biological replicates ($n = 3$); consensus values are reported.
(TIF)

**S8 Fig. Heatmap of antimicrobial potency of SLS.** Heatmap showing MIC, MBC, MBIC, and MBEC values (mg/mL) of SLS against the tested pathogens. Lower values indicate higher antimicrobial potency. All assays were performed in three independent biological replicates ($n = 3$); consensus values are reported.
(TIF)

**S9 Fig. Box plot of zone-of-inhibition values for TM, SA, and SLS.** Box plots summarizing the distribution of inhibition zone diameters (mm) for TM, SA, and SLS across all tested pathogens. Statistical comparisons were performed using one-way ANOVA followed by Tukey's HSD post-hoc test. Data represent three independent biological replicates ($n = 3$).
(TIF)

**S10 Fig. Violin plot of zone-of-inhibition values for TM, SA, and SLS.** Violin plots illustrating the variation and density distribution of inhibition zone diameters (mm) for TM, SA, and SLS. Statistical analyses were performed using one-way ANOVA with Tukey's HSD post-hoc test. Data represent three independent biological replicates ($n = 3$).
(TIF)

**S11 Fig. Stability and binding analysis of the 5CXK–TM complex during a 300 ns MD simulation.** This figure presents RMSD, RMSF, radius of gyration (Rg), solvent-accessible surface area (SASA), hydrogen-bond profiles, and MM/GBSA binding free energy ($\Delta G$) for the 5CXK–TM complex. Estimated $K_d$ and $IC_{50}$ values derived from $\Delta G$ are also included, summarizing overall complex stability and binding strength across 300 ns.
(TIF)

**S12 Fig. Stability and binding analysis of the 4DXD–TM complex during a 300 ns MD simulation.** RMSD, RMSF, Rg, SASA, hydrogen-bond profiles, and MM/GBSA binding free energy ($\Delta G$) for the 4DXD–TM complex are shown, along with estimated $K_d$ and $IC_{50}$ values, providing a summary of structural stability and binding behavior over 300 ns.
(TIF)

**S13 Fig. Stability and binding analysis of the 5TZ1–TM complex during a 300 ns MD simulation.** This figure includes RMSD, RMSF, Rg, SASA, hydrogen-bond patterns, and MM/GBSA binding free energy ($\Delta G$) for the 5TZ1–TM complex, with corresponding $K_d$ and $IC_{50}$ estimates reflecting overall stability and binding affinity across 300 ns.
(TIF)

**S14 Fig. Stability and binding analysis of the 5CXK–PA complex during a 300 ns MD simulation.** RMSD, RMSF, Rg, SASA, hydrogen-bond profiles, and MM/GBSA binding free energy ($\Delta G$) for the 5CXK–PA complex are presented, together with estimated $K_d$ and $IC_{50}$ values that summarize binding strength and structural stability throughout 300 ns.
(TIF)

**S15 Fig. Stability and binding analysis of the 4URN–PA complex during a 300 ns MD simulation.** This figure reports RMSD, RMSF, Rg, SASA, hydrogen-bond data, and MM/GBSA binding free energy ($\Delta G$) for the 4URN–PA complex, along with derived $K_d$ and $IC_{50}$ values, providing insight into complex stability over 300 ns.
(TIF)

**S16 Fig. Stability and binding analysis of the 5TZ1–PA complex during a 300 ns MD simulation.** RMSD, RMSF, Rg, SASA, hydrogen-bond profiles, and MM/GBSA binding free energy ($\Delta G$) are shown for the 5TZ1–PA complex. Estimated $K_d$ and $IC_{50}$ values further summarize binding affinity and stability during 300 ns.
(TIF)

**S17 Fig. Stability and binding analysis of the 1LMH–LS complex during a 300 ns MD simulation.** This figure presents RMSD, RMSF, Rg, SASA, hydrogen-bond behavior, and MM/GBSA binding free energy ($\Delta G$) for the 1LMH–LS complex, with estimated $K_d$ and $IC_{50}$ values indicating overall complex stability and binding strength across 300 ns.
(TIF)

**S18 Fig. Stability and binding analysis of the 2VES–LS complex during a 300 ns MD simulation.** RMSD, RMSF, Rg, SASA, hydrogen-bond profiles, and MM/GBSA binding free energy ($\Delta G$) for the 2VES–LS complex are included, along with $K_d$ and $IC_{50}$ estimates summarizing stability and binding interactions during the whole 300 ns simulation.
(TIF)

**S19 Fig. Stability and binding analysis of the 5JPF–LS complex during a 300 ns MD simulation.** This figure shows RMSD, RMSF, Rg, SASA, hydrogen-bond trends, and MM/GBSA binding free energy ($\Delta G$) for the 5JPF–LS complex. Estimated $K_d$ and $IC_{50}$ values provide an overview of binding affinity and structural stability over 300 ns.
(TIF)

**S20 Fig. PCA of the 5CXK–TM complex. (A)** PC1 and PC2 projections over 300 ns showing large-amplitude fluctuations and a major conformational transition near 140 ns, followed by stabilization. **(B)** Free-energy landscape (FEL) along PC1–PC2, revealing multiple shallow to moderately deep minima (~0–15 kJ/mol), indicating broad conformational sampling. **(C)**

PC1–PC2 scatter plot showing wide dispersion of conformations. **(D)** Scree plot demonstrating that PC1 and PC2 together explain ~80–85% of the total variance, indicating dominance of a few collective motions.
(TIF)

**S21 Fig. PCA of the 4DXD–TM complex. (A)** PC1 and PC2 vs time showing low-amplitude, restricted fluctuations, consistent with high structural stability. **(B)** FEL indicates two compact deep minima (~0–8 kJ/mol), representing well-defined stable states. **(C)** PC1–PC2 scatter plot showing dense clustering within limited regions. **(D)** Scree plot showing that PC1 and PC2 account for ~60–65% of variance, suggesting additional small modes contribute to residual motion.
(TIF)

**S22 Fig. PCA of the 5TZ1–TM complex. (A)** PC1–PC2 projection showing moderate fluctuations and periodic transitions throughout the simulation. **(B)** FEL displaying several shallow minima (~0–7 kJ/mol), suggesting flexible yet stable conformational states. **(C)** PC1–PC2 scatter plot showing moderate dispersion. **(D)** Scree plot indicating that PC1 and PC2 represent ~70% of the total variance, reflecting intermediate essential dynamics.
(TIF)

**S23 Fig. PCA of the 4URN–PA complex. (A)** PC1 and PC2 vs time showing very restricted fluctuations (PC1 SD±0.43; PC2 SD±0.36) with minimal transitions. **(B)** FEL showing a single compact shallow basin (~0–3 kJ/mol), indicating strong conformational stability. **(C)** PC1–PC2 scatter plot showing narrow, dense clustering. **(D)** Scree plot showing variance spread across several low-amplitude modes, indicating the absence of dominant motions.
(TIF)

**S24 Fig. PCA of the 5CXK–PA complex. (A)** PC1–PC2 projection showing large-amplitude transitions during the first 50 ns, followed by stabilization (PC1 SD±4.91; PC2 SD±1.66). **(B)** FEL with two moderately deep minima (~0–12 kJ/mol), corresponding to metastable conformational states. **(C)** Scatter plot showing broad dispersion dominated by PC1-driven motions. **(D)** Scree plot indicating that PC1 accounts for most of the variance, reflecting strong collective motions.
(TIF)

**S25 Fig. PCA of the 5TZ1–PA complex. (A)** PC1 and PC2 vs time showing moderate fluctuations (PC1 SD±1.48; PC2 SD±1.06) with periodic transitions. **(B)** FEL containing several shallow minima (~0–8 kJ/mol), indicating balanced flexibility and stabilization. **(C)** PC1–PC2 scatter plot showing intermediate sampling—broader than 4URN–PA but narrower than 5CXK–PA. **(D)** Scree plot indicating a moderate distribution of variance across principal components.
(TIF)

**S26 Fig. PCA of the 1LMH–LS complex. (A)** PC1–PC2 vs time showing gradual moderate fluctuations (PC1 SD±1.48; PC2 SD±1.06). **(B)** FEL with multiple shallow minima (~0–6 kJ/mol), consistent with flexible but stable states. **(C)** Scatter plot showing dispersed sampling across PCs. **(D)** Scree plot indicating dominance of the first few modes.
(TIF)

**S27 Fig. PCA of the 2VES–LS complex. (A)** PC1 and PC2 vs time showing highly restricted motion (PC1 SD±0.67; PC2 SD±0.47). **(B)** FEL showing two compact low-energy minima (~0–5 kJ/mol), confirming strong conformational stabilization. **(C)** Scatter plot showing tight clustering into two major conformational states. **(D)** Scree plot showing variance distributed across low-amplitude modes, indicating constrained dynamics.
(TIF)

**S28 Fig. PCA of the 5JPF–LS complex. (A)** PC1–PC2 projection showing intermediate fluctuations (PC1 SD±0.85; PC2 SD±0.67) with frequent transitions. **(B)** FEL displaying several moderately deep minima (~0–7 kJ/mol), reflecting a broader conformational landscape. **(C)** PC1–PC2 scatter plot showing wider sampling than 2VES–LS but more restricted

than 1LMH–LS. **(D)** Scree plot showing steep eigenvalue decay, indicating that essential motions are captured in the early PCs.
(TIF)

**S29 Fig. Heatmap of predicted high-risk toxicities for TM.** Heatmap showing predicted probabilities of hepatotoxicity, carcinogenicity, genotoxicity, neurotoxicity, and nephrotoxicity for TM. Dark red indicates higher predicted toxicity (values approaching 1), while lighter shades indicate lower risk.
(TIF)

**S30 Fig. Heatmap of the predicted high-risk toxicities for SA.** Heatmap illustrating predicted probabilities of toxicity across hepatotoxicity, carcinogenicity, genotoxicity, neurotoxicity, and nephrotoxicity for SA. Color intensity reflects the likelihood of adverse effects.
(TIF)

**S31 Fig. Heatmap of predicted high-risk toxicities for SLS.** Heatmap displaying predicted toxicity probabilities for SLS across hepatotoxicity, carcinogenicity, genotoxicity, neurotoxicity, and nephrotoxicity. A darker color corresponds to a higher predicted risk.
(TIF)

**S1 Table. Protein targets selected for *in silico* molecular docking.** This table lists the microbial proteins screened against thymol (TM), phenyl azide (PA; analog of sodium azide [SA]), and lauryl sulfate (LS; an analog of sodium lauryl sulfate [SLS]). For each target, the enzyme name, source organism, FASTA sequence, amino acid length, structural information (PDB code or model source), and associated biological pathway are provided. These proteins represent essential bacterial and fungal pathways and were selected to enable a comprehensive evaluation of ligand–protein interactions.
(DOCX)

**S2 Table. Structural quality metrics of AlphaFold-predicted models of DabA (MpsA) and DabB (MpsB).** This table summarizes the structural validation parameters for the AlphaFold-predicted models of DabA and DabB, including the total number of residues, Rg, mean and median pLDDT scores, and the proportion of residues within high-confidence (≥90), confident (70–89), and low-confidence (<50) ranges. Both proteins exhibit compact folds and high overall prediction confidence, with mean pLDDT scores of 87.92 for DabA and 88.69 for DabB and minimal low-confidence regions. These metrics support the suitability of the models for downstream molecular docking and molecular dynamics simulations.
(DOCX)

## Acknowledgments

The Researchers would like to thank the Deanship of Graduate Studies and Scientific Research at Qassim University for financial support (QU-APC-2026).

## Author contributions

**Conceptualization:** Kamal Ahmad Qureshi.

**Data curation:** Kamal Ahmad Qureshi.

**Formal analysis:** Kamal Ahmad Qureshi, Adil Parvez.

**Funding acquisition:** Kamal Ahmad Qureshi.

**Investigation:** Kamal Ahmad Qureshi, Adil Parvez.

**Methodology:** Kamal Ahmad Qureshi.

**Project administration:** Kamal Ahmad Qureshi.

**Resources:** Kamal Ahmad Qureshi.

**Software:** Kamal Ahmad Qureshi.

**Supervision:** Kamal Ahmad Qureshi.

**Validation:** Kamal Ahmad Qureshi, Adil Parvez.

**Visualization:** Kamal Ahmad Qureshi.

**Writing – original draft:** Kamal Ahmad Qureshi.

**Writing – review & editing:** Kamal Ahmad Qureshi, Adil Parvez.

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
