## [Decision Letter · Decision Letter 0]

5 Nov 2025

Dear Dr. Qureshi,

Thank you for submitting your manuscript to PLOS ONE. After careful consideration, we feel that it has merit but does not fully meet PLOS ONE’s publication criteria as it currently stands. Therefore, we invite you to submit a revised version of the manuscript that addresses the points raised during the review process.

We look forward to receiving your revised manuscript.

Kind regards,

Jorddy Neves Cruz

Academic Editor

PLOS ONE

Journal Requirements:

“The Researchers would like to thank the Deanship of Graduate Studies and Scientific Research at Qassim University for their financial support (QU-APC-2025).”

3. We note that your Data Availability Statement is currently as follows: All relevant data are within the manuscript and in Supporting Information files.

Reviewers' comments:

Reviewer's Responses to Questions

**Comments to the Author**

1. Is the manuscript technically sound, and do the data support the conclusions?

Reviewer #1: Yes

Reviewer #2: Yes

Reviewer #3: Partly

Reviewer #4: Yes

2. Has the statistical analysis been performed appropriately and rigorously?

Reviewer #1: Yes

Reviewer #2: Yes

Reviewer #3: I Don't Know

Reviewer #4: No

3. Have the authors made all data underlying the findings in their manuscript fully available?

Reviewer #1: No

Reviewer #2: Yes

Reviewer #3: No

Reviewer #4: Yes

4. Is the manuscript presented in an intelligible fashion and written in standard English?

Reviewer #1: Yes

Reviewer #2: Yes

Reviewer #3: Yes

Reviewer #4: Yes

Reviewer #1: 1. Scientific terms such as in vitro, in vivo, and in silico should be italicized throughout the manuscript. Please check and revise accordingly.

2. The images currently provided have colored backgrounds. It would be preferable to present them with a plain (white or transparent) background for better clarity and uniformity.

3 Please verify whether AlphaFold structures are available for the MpsA and MpsB proteins. If available, use these structures for the computational analyses.

4. A detailed molecular dynamics (MD) simulation protocol should be provided, including the parameters used for energy minimization, equilibration, and production phases.

5. Clearly specify whether the MM-GBSA or MM-PBSA method was used for binding free energy calculations.

6. Table 1 should be expanded to include hydrogen-bonding residues along with their bond distances. Additionally, check for possible π–π stacking interactions and include them in the table if present.

7. Certain parts of the Results section are presented as bullet points. Please rewrite these sections in paragraph form for improved readability.

8. Some words in the Results section are highlighted in bold. Please remove the bold formatting.

9. In Lines 388–390, you state:

“RMSF analyses (Figs. 17–19B) showed reduced fluctuations in active-site residues for 4URN–PA and 5TZ1–PA, highlighting the ability of PA to stabilize key residues, particularly those involved in hydrogen bonding and hydrophobic interactions.”

Please visually highlight the reduced fluctuations of active-site residues in the RMSF plot for clarity.

10. The Results and Discussion section on MD simulations lacks sufficient statistical detail. Please include information such as, Equilibration time and pattern observed, Residues showing higher flexibility, and Average number of hydrogen bonds maintained during the simulation

11. The discussion on PCA-based FEL (Free Energy Landscape) analysis is insufficient. Please provide a more detailed explanation of the FEL plots in terms of conformational stability and energy minima of the complexes.

12. Authors are requested to deposit their MD simulation trajectories and associated data in a publicly accessible repository (e.g., Zenodo, Figshare, or Mendeley Data) and provide the corresponding access link within the manuscript for transparency and reproducibility.

Reviewer #2: The manuscript addresses a timely and important issue by exploring alternative antimicrobial agents against multidrug-resistant (MDR) pathogens, with a particular focus on thymol (TM). The integration of in vitro assays with in silico modeling, especially molecular docking, MD simulations, and ADMET profiling, strengthens the study's scientific rigor and provides a comprehensive evaluation of TM’s therapeutic potential. My comments and concerns are given below.

• The manuscript states that an RMSD ≤ 2.0 Å was used as the criterion for successful re-docking validation. Could the authors elaborate on why the goal was not to achieve a 0 Å RMSD, or clarify the rationale behind choosing the 2.0 Å threshold instead?

• The authors mention that the structures of MpsA and MpsB were modeled using SWISS-MODEL. Could the authors clarify why AlphaFold, which generally provides higher-accuracy structural predictions, was not used as an alternative or for comparison?

• The manuscript currently includes a large number of figures in the main text. To improve readability and streamline the presentation, it is recommended to reduce the number of figures in the main manuscript to a maximum of 12. The remaining figures can be moved to the supplementary material.

Reviewer #3: The study screens thymol (TM), sodium azide (SA) and sodium lauryl sulfate (SLS) against a broad panel of bacteria and two fungi, reports disc-diffusion/MIC/MBC/MBIC/MBEC outcomes, and complements these with docking, 300-ns MD, MM/GBSA, PCA/FEL and ADMET predictions.

Major concerns -

1. The title need to be revised. There are repetitions such as in silico and computational, in vitro and the assays stated! Shorten the title to a sentence that covers the whole study.

2. The compounds TM, SA, SLS, PA, LS should be fully abbreviated when mentioned in the first place.

3. The introduction is weak. The background leading to the development of hypothesis is lacking. Weak contextualization of the problem and the flow of discussion is also missing. It reads more like a descriptive summary of antimicrobial resistance rather than a tightly argued justification for why this specific combination of compounds and approaches were used in this study.

4. The novelty of the work is not highlighted.

5. The antimicrobial activity of thymol is well studied. How this study is different from the previous studies should be justified. The authors can refer to the following articles-

a) https://www.frontiersin.org/journals/microbiology/articles/10.3389/fmicb.2020.01744/full

b) https://pmc.ncbi.nlm.nih.gov/articles/PMC9870694/

6. It is not clear and why SA and SLS were chosen to benchmark TM? No standard drug/antibiotic was used in the experiment! justify their inclusion versus clinically relevant comparators.

7. The author used clinical isolate. Was it from microbial bank or direct from patient? The Ethical approval is not provided. If not required then explain why approval was not required.

8. For statistical analysis, the number of replicates should be mentioned. Only biological replicate is acceptable but not the technical replicate.

9. What was the final concentration of DMSO? State in the method.

10. The quality of the figures should be improved substantially.

11. Protein 3D structure modeling method together with validation should be given if not in the manuscript but as supplement.

12. The conclusion is too broad. Focus the significant finding and the carry out message with future direction.

Reviewer #4: This manuscript presents a well-structured and ambitious study combining both in-vitro and in-silico approaches to evaluate thymol (TM), sodium azide (SA), and sodium lauryl sulfate (SLS) as antimicrobial and antibiofilm agents. The integration of biochemical assays with molecular docking, molecular dynamics (MD) simulations, and ADMET profiling provides a multi-dimensional analysis that is commendable and scientifically relevant.

The work is timely given the ongoing search for new strategiesto combat multidrug-resistant (MDR) pathogens. However, while the study demonstrates strong technical execution and clear motivation, certain aspects of experimental design, computationalinterpretation, and data presentation would benefit from refinement and more critical discussion.

The combined use of in-vitro and in-silico methods is appropriate and adds value by linking observed antimicrobial activity to predicted molecular interactions.

1. The study’s focus on repurposing thymol and eevaluating structural analogs (e.g., LS) as potential scaffolds is a meaningful contribution to antimicrobial drug discovery. However, the novelty is moderate thymol’s antimicrobial potential is already well-documented, and the computational exploration of its binding to common microbial targetes has been reported in prior literature. The manuscript would benefit from a clearer articulation of what specifically is new about this study (e.g., the particular combination of targets, unique comparative approach, or mechanistic insight)

2. The in-vitro assays (MIC, MBC, and biofilm inhibition) are described adequately, but it is not always clear how many biological or technical replicates were performed. Including these details would enhance reproducibility.

3. The selection of control agents and bacterial strains seems logical, though some justification for their choice would help (e.g., why these particular MDR species were prioritized).

4. The computational workflow (molecular docking MD simulations, MM-GBSA, PCA/FEL) is methodologically robust. Yet, the assumed correlation between ΔG and IC₅₀ could be discussed more cautiously; this relationship is non-linear and depends heavily on experimental context.

5. While 300-ns MD simulations are impressive, the analysis would be strengthened by including convergence checks such as clustering or RMSD stability analysis beyond PCA/FEL plots.

6. The integration of in-vitro and in-silico data is one of the paper’s strongest features, clearly linking molecular behavior with biological activity. However, the presentation could be streamlined: several figures and tables seem to repeat similar information (e.g., multiple mentions of binding energies or RMSD trends). Condensing these could improve readability.

7. The discussion of SA and SLS is appropriately cautious, acknowledging their toxicity and poor pharmacokinetics. Still, a deeper reflection on why these compounds were chosen beyond being “comparative controls” would clarify their scientific role.

8. Nonetheless, the analysis occasionally mixes predicted and literature-derived toxicities without clearly separating computational data from empirical evidence. Labeling which parameters were experimentally validated versus purely predicted would enhance transparency.

9. The language could be tightened certain sections repeat the same point (e.g., TM’s superiority) multiple times.

10. The connection to clinical translation could be expanded: what specific formulations or combination therapies are most feasible for TM?

11. The paper would benefit from a graphical summary or schematic showing the overall workflow (in-vitro testing docking,MD, ADMET, outcome).

12. Technical terms are used correctly, though some sentences are overly long or complex. Simplifying these would make the text more reader-friendly.

13. Figures (especially PCA/FEL plots) should include better labeling and resolution, with consistent font sizes and color schemes to enhance readability.

14. Explicitly state what new insight this study provides beyond existing thymol literature.

15. Include replicate numbers, control information, and statistical tests.

16. Address limitations of Ki ≈ Kd assumptions and possible docking bias.

17. Distinguish predicted vs. empirical ADMET findings.

18. Improve figure quality and legends and remove the back ground colors from the figures (must needed)

.

Reviewer #1: No

Reviewer #2: No

Reviewer #3: **Yes:** AKM Moyeenul HuqAKM Moyeenul HuqAKM Moyeenul HuqAKM Moyeenul Huq

Reviewer #4: No

---

## [Author Response · Author response to Decision Letter 1]

6 Dec 2025

Detailed responses to all reviewers’ comments have been submitted as a separate PDF document.

---

## [Decision Letter · Decision Letter 1]

21 Dec 2025

Dear Dr. Qureshi,

Thank you for submitting your manuscript to PLOS ONE. After careful consideration, we feel that it has merit but does not fully meet PLOS ONE’s publication criteria as it currently stands. Therefore, we invite you to submit a revised version of the manuscript that addresses the points raised during the review process.

We look forward to receiving your revised manuscript.

Kind regards,

Jorddy Neves Cruz

Academic Editor

PLOS One

**Journal Requirements:**

Reviewers' comments:

Reviewer's Responses to Questions

**Comments to the Author**

Reviewer #5: (No Response)

Reviewer #6: All comments have been addressed

2. Is the manuscript technically sound, and do the data support the conclusions?

Reviewer #5: Yes

Reviewer #6: Yes

3. Has the statistical analysis been performed appropriately and rigorously?

Reviewer #5: No

Reviewer #6: Yes

4. Have the authors made all data underlying the findings in their manuscript fully available?

Reviewer #5: Yes

Reviewer #6: Yes

5. Is the manuscript presented in an intelligible fashion and written in standard English?

Reviewer #5: Yes

Reviewer #6: Yes

Reviewer #5: In the submitted manuscript entitled “Antimicrobial and Antibiofilm Evaluation of Thymol, Sodium Azide, and Sodium Lauryl Sulfate Against Multidrug-Resistant Pathogens: An Integrated Experimental and Computational Study”, the authors integrates in vitro antimicrobial and antibiofilm assays with comprehensive in silico analyses to evaluate the repurposing potential of thymol (TM), sodium azide (SA), and sodium lauryl sulfate (SLS) against bacterial and 2 fungal strains, including MRSA clinical isolates. As well, the authors performed molecular docking computations and molecular dynamics simulations over 300 ns. The manuscript is well-presented, and the results are well-discussed. Further technical details regarding the molecular docking calculations must be presented. Full technical details regarding molecular docking computations and molecular dynamics simulations should be provided in the revised manuscript. All figures are poorly presented. Consequently, the manuscript in its current form is not suitable for publication in PLOS ONE.

Comments:

#########

Comment #0: The manuscript must be revised, where some typos and grammatical errors were observed.

Comment #1: The introduction provides a general context for the study. However, it does not explain the premises of the work, nor does it provide a more specific rationale for the study. Additionally, the results should not be presented in the introduction section.

Comment #2: Could the authors clarify the rationale for selecting this older PDB entry, despite the availability of more recent structures with improved resolution?

Comment #3: 1000 ps is not sufficient for the equilibration stage of the investigated complexes nowadays. It should be at least 10 ns.

Comment #4: The authors claimed that “the ADMET evaluation was conducted using SwissADME to assess the drug-likeness properties of the selected compounds”. However, there is a big difference between ADMET and drug likeness properties. Please revise and correct.

Comment #5: Full technical details regarding molecular docking and molecular dynamics simulations should be provided in the revised manuscript.

Comment #6: How did the authors compute the atomic charges of the investigated compounds prior to molecular dynamics simulations?

Comment #7: How did the authors compute the atomic charges of the investigated compounds prior to molecular docking computations?

Comment #8: Figures 7 to 12 should be combined into one Figure.

Comment #9: Molecular interactions of the predicted docking poses of the identified analogs should be detailed in the revised manuscript.

Comment #10: Post-MD calculations should include binding energy per frame, center-of-mass, and hydrogen bond analyses..., etc.

Comment #11: Reference style must be revised. There are versatile omitted details.

Comment #12: The conclusion section must be rewritten to be more informative and unveil the most beneficial outcomes.

Comment #13: All docking scores and binding energies should be in one decimal unit.

Comment #14: Did the authors investigate the protonation state of the titrable amino acids before docking?

Comment #15: Did the authors investigate the protonation state of the ligands before docking computations?

Comment #16: Full technique details regarding molecular docking computations should be considered in the revised manuscript, such as the grid spacing value.

Comment #17: The reference of the utilized pdb code should be cited in the revised manuscript.

Comment #18: Did the authors perform equilibration for the investigated complexes prior to executing molecular dynamics simulations?

Reviewer #6: Reviewer’s Comment

Manuscript Number: PONE-D-25-53573R1

Title: Antimicrobial and Antibiofilm Evaluation of Thymol, Sodium Azide, and Sodium Lauryl Sulfate Against Multidrug-Resistant Pathogens: An Integrated Experimental and Computational Study

This manuscript presents a timely and methodologically robust investigation into the repurposing potential of three compounds against multidrug-resistant pathogens. The study effectively bridges experimental microbiology with advanced computational modeling, addressing a critical need for novel antimicrobial strategies, particularly against biofilms. The work aligns well with PLOS ONE's scope of publishing rigorous, multidisciplinary scientific research.

Strengths Supporting Acceptance:

1. Significant and Relevant Topic: The research tackles the pressing global health challenge of antimicrobial resistance (AMR) and biofilm-associated infections, a core interest area for the scientific and medical community.

2. Comprehensive and Integrated Methodology: The combination of in vitro assays (MIC, MBC, MBIC, MBEC) across a broad panel of clinically relevant bacterial and fungal strains with extensive in silico analyses (molecular docking, 300 ns MD simulations, MM-GBSA, PCA/FEL, ADMET) represents a notable strength. This multi-pronged approach provides a more complete picture of the compounds' potential than either approach alone.

3. Clear and Compelling Findings: The abstract succinctly presents differentiated roles for the tested agents:

• Thymol (TM) is identified as a strong, broad-spectrum repurposing candidate with dual antimicrobial/antibiofilm activity and a favorable predicted safety profile.

• The study intelligently uses SA and SLS as mechanistic benchmarks, providing context for the novel findings.

• The in silico exploration of non-ionic analogs (PA, LS) as optimized scaffolds, particularly the high predicted affinity of LS for LpxC, generates a valuable hypothesis for future drug development.

4. Rigorous Computational Analysis: The use of long-timescale MD simulations (300 ns), binding free energy calculations (MM-GBSA), and conformational analysis (PCA/FEL) moves beyond simple docking to provide credible insights into binding stability and dynamics, strengthening the mechanistic hypotheses.

5. Translational Perspective: The focus on "repurposing potential" and the inclusion of in silico ADMET profiling directly address the practical hurdles in antimicrobial development, increasing the translational relevance of the work.

Overall Assessment:

I recommend an acceptance because this is a well-conceived and expertly executed study that makes a valuable contribution to the field of AMR research. The integration of complementary techniques strengthens the conclusions and provides a clear pathway for future work. The manuscript is likely to be of broad interest to microbiologists, pharmacologists, and computational biologists.

.

Reviewer #5: No

Reviewer #6: **Yes:** Olayinka Oyewale AJANIOlayinka Oyewale AJANIOlayinka Oyewale AJANIOlayinka Oyewale AJANI

---

## [Author Response · Author response to Decision Letter 2]

31 Jan 2026

A detailed, point-by-point response to Reviewer-5's comments has been uploaded as a separate PDF for ease of review.

---

## [Decision Letter · Decision Letter 2]

12 Mar 2026

Antimicrobial and Antibiofilm Evaluation of Thymol, Sodium Azide, and Sodium Lauryl Sulfate Against Multidrug-Resistant Pathogens: An Integrated Experimental and Computational Study

PONE-D-25-53573R2

Dear Dr. Qureshi,

We’re pleased to inform you that your manuscript has been judged scientifically suitable for publication and will be formally accepted for publication once it meets all outstanding technical requirements.

Kind regards,

Jorddy Neves Cruz

Academic Editor

PLOS One

Additional Editor Comments (optional):

Reviewers' comments:

Reviewer's Responses to Questions

**Comments to the Author**

Reviewer #5: All comments have been addressed

2. Is the manuscript technically sound, and do the data support the conclusions?

Reviewer #5: Yes

3. Has the statistical analysis been performed appropriately and rigorously?

Reviewer #5: Yes

4. Have the authors made all data underlying the findings in their manuscript fully available?

Reviewer #5: Yes

5. Is the manuscript presented in an intelligible fashion and written in standard English?

Reviewer #5: Yes

Reviewer #5: In the revised manuscript entitled "Antimicrobial and Antibiofilm Evaluation of Thymol, Sodium Azide, and Sodium Lauryl Sulfate Against Multidrug-Resistant Pathogens: An Integrated Experimental and Computational Study", the authors considered all our comments. Consequently, the manuscript may be published in the Journal of Plos one.

.

Reviewer #5: No

---

## [Editor Report · Acceptance letter]

PONE-D-25-53573R2

PLOS One

Dear Dr. Qureshi,

I'm pleased to inform you that your manuscript has been deemed suitable for publication in PLOS One. Congratulations! Your manuscript is now being handed over to our production team.

Kind regards,

on behalf of

Dr. Jorddy Neves Cruz

Academic Editor

PLOS One